# ON THE INTERACTION OF COMPRESSIBILITY AND ADVERSARIAL ROBUSTNESS

**Melih Barsbey**
Department of Computing
Imperial College London, UK

**Antônio H. Ribeiro**
Department of Information Technology
Uppsala University, Sweden

**Umut Şimşekli**
INRIA, CNRS, Département d'Informatique
ENS / PSL, France

**Tolga Birdal**
Department of Computing
Imperial College London, UK

## ABSTRACT

As demands for resource efficiency and safety in modern neural networks intensify, substantial research effort has gone into model compression and adversarial robustness. Yet despite progress on each in isolation, a systematic understanding of how compressibility shapes robustness remains elusive. In this paper, we develop a principled framework to analyze how different forms of structured compressibility - such as neuron-level and spectral compressibility - affect adversarial robustness. We show that structured compressibility can induce a small number of highly sensitive directions in the representation space, which adversaries can exploit to construct effective perturbations. Our analysis yields a robustness bound that reveals how neuron and spectral compressibility impact $\ell_\infty$ and $\ell_2$ robustness via their effects on the learned representations. Crucially, the vulnerabilities we identify arise irrespective of how compressibility is achieved - whether via regularization, architectural bias, or learning dynamics. Through empirical evaluations across synthetic and realistic tasks, we confirm our theoretical predictions, and further demonstrate that these vulnerabilities persist under adversarial training and transfer learning, and contribute to the emergence of universal adversarial examples. Our findings show a fundamental tension between structured compressibility and robustness and highlight new pathways for designing models that are efficient and safe.

## 1 INTRODUCTION

Machine learning systems are increasingly deployed in safety-critical domains such as healthcare (Rajpurkar et al., 2022) and autonomous driving (Hussain & Zeadally, 2019), where reliability is paramount. With their growing social impact, modern neural networks are now expected to not only be more resource-efficient, e.g. amenable to compression in favor of reduced memory footprint and latency, but to do so while retaining their safety properties. In this paper, we focus on a primary safety concern in adversarial robustness, and investigate how it interacts with compressibility. While both topics have been studied extensively in isolation, a mature and unified understanding of how compressibility shapes adversarial robustness is still lacking.

As desirable as adversarial robustness and compressibility both are, the research has been equivocal regarding whether/when/how their simultaneous achievement is possible (Guo et al., 2018; Balda et al., 2020; Li et al., 2020a; Merkle et al., 2022; Liao et al., 2022). This is even more pronounced for *structured* compressibility, which is alarming given its practical relevance (Blalock et al., 2020; Piras et al., 2025). However, recent research has started to provide mechanism-based explanations for this relationship, highlighting how compressibility impacts models' vulnerability to adversarial noise. For example, Savostianova et al. (2023) demonstrate that low-rank parameterizations may inadvertently amplify local Lipschitz constants, increasing sensitivity to perturbations. Nern et al. (2023) connect adversarial transferability to layer-wise operator norms and their impact on representation geometry. Feng et al. (2025) further show that while moderate sparsity can enhance robustness, excessive sparsity causes ill-conditioning that reintroduces fragility and vulnerability. These results hint at a

Figure 1: A visual preview of our findings. (Left) Sparsification expedites compression but creates sensitive latent directions. (Center) Adversaries exploit these sensitive directions to increase their potency. (Right) This leads to decreased adversarial robustness.

delicate, regime-dependent relationship between compressibility and robustness - but a principled and general framework is still lacking.

In this work, we develop a framework to investigate the effect of structured compressibility on adversarial robustness through its effect on parameter operator norms and network's Lipschitz constant. We jointly study how different forms of compressibility - particularly neuron-level and spectral compressibility - affect adversarial robustness. Our central result is an instructive adversarial robustness bound that reveals how compressibility can lead to adversarial vulnerability by inducing a small set of highly sensitive directions in the representation space. Empirically, we confirm this effect across architectures, datasets, and attack models, where adversarial attacks reliably identify and exploit these sensitive directions. Figure 1 provides a visual preview of our findings. Previous research tightly links compressibility to generalization (Arora et al., 2018; Barsbey et al., 2021); however, our findings imply that the very mechanisms that promote generalization can also introduce structural weaknesses. In summary, our contributions are:

1. We provide an **adversarial robustness bound** that decomposes into analytically interpretable terms, and predicts that neuron and spectral compressibility create adversarial vulnerability against $\ell_\infty$ and $\ell_2$ attacks, through their effects on networks' Lipschitz constants.
2. Utilizing various compressibility-inducing interventions, we empirically validate our predictions regarding the **emergence of adversarial vulnerability under structured compressibility** with various datasets and models, including commonly used modern encoder architectures.
3. We demonstrate that the **detrimental effects of compressibility persist under adversarial training and transfer learning**, and contribute to the appearance of universal adversarial examples.
4. We demonstrate and discuss our findings' implications for compression in practice, and highlight promising paths for **designing models that reconcile efficiency and safety**.

We provide our source code at (`https://github.com/mbarsbey/advcomp`).

## 2 SETUP

**Notation**. We denote scalars by lower case italic ($k$), vectors with lower case bold ($\boldsymbol{x}$), and matrices with upper case bold ($\mathbf{W}$) characters respectively. Vector $\ell_p$ norms are denoted by $\|\boldsymbol{x}\|_p$. For matrices, $\|\mathbf{W}\|_F, \|\mathbf{W}\|_2, \|\mathbf{W}\|_\infty$ correspond to Frobenius, spectral, and $\ell_\infty$-$\ell_\infty$ operator norms, respectively. We denote the $i^{\text{th}}$ element of a vector $\boldsymbol{x}$ with $x_i$, and row $i$ of a matrix $\mathbf{W}$ with $\mathbf{w}_i$. Elements of a sequence of matrices (e.g. layer matrices) are referred to by $\mathbf{W}^l, l \in [\lambda]$, where for $\lambda \in \mathbb{N}$ we let $[\lambda] := \{1, \ldots, \lambda\}$. Unless otherwise specified, we will be focusing on supervised classification problems, which will involve the input $\boldsymbol{x} \in \mathcal{X}$ and label $y \in \mathcal{Y}$. A predictor $g : \mathcal{X} \to \mathbb{R}^{|\mathcal{Y}|}$, parametrized by $\boldsymbol{\theta} \in \Theta$ produces output logits $\boldsymbol{s} = g(\boldsymbol{x}, \boldsymbol{\theta})$, the maximum of which is the predicted label $\hat{y} = \arg\max_{i \in [|\mathcal{Y}|]} s_i$. Predictions are evaluated by a loss function $\ell : \mathbb{R}^{|\mathcal{Y}|} \times \mathcal{Y} \to \mathbb{R}_+$. For brevity, we define the composite loss function $f(\boldsymbol{x}, \boldsymbol{\theta}) := \ell(g(\boldsymbol{x}, \boldsymbol{\theta}), y)$.

**Risk and adversarial robustness**. Assuming a data distribution $\pi$ on $\mathcal{X} \times \mathcal{Y}$, we define the population and empirical risks as $F(\boldsymbol{\theta}) := \mathbb{E}_{\boldsymbol{x}, y \sim \pi}[f(\boldsymbol{x}, \boldsymbol{\theta})]$, and $\widehat{F}(\boldsymbol{\theta}, S) := \frac{1}{n} \sum_{i=1}^n f(\boldsymbol{x}_i, \boldsymbol{\theta})$, where $(\boldsymbol{x}_i, y_i)_{i=1}^n$ denotes a set of i.i.d. samples from $\pi$. Adversarial attacks are small perturbations to input that dramatically disrupt a model's predictions (Szegedy et al., 2014). In this paper, we focus on bounded $p$-norm attacks, which we define as

$$\boldsymbol{a}^* = \arg\max_{\|\boldsymbol{a}\|_p \leq \delta} f(\boldsymbol{x} + \boldsymbol{a}, \boldsymbol{\theta}). \tag{1}$$

Given the adversarial loss $f_p^{\mathrm{adv}}(\boldsymbol{x}, \boldsymbol{\theta}; \delta) := f(\boldsymbol{x} + \boldsymbol{a}^*, \boldsymbol{\theta})$, we define adversarial risk and empirical adversarial risk as $F_p^{\mathrm{adv}}(\boldsymbol{\theta}; \delta) := \mathbb{E}_{\boldsymbol{x}, y \sim \pi}[f_p^{\mathrm{adv}}(\boldsymbol{x}, \boldsymbol{\theta}; \delta)]$ and $\widehat{F}_p^{\mathrm{adv}}(\boldsymbol{\theta}, S; \delta) := \frac{1}{n} \sum_{i=1}^{n} f_p^{\mathrm{adv}}(\boldsymbol{x}_i, \boldsymbol{\theta}; \delta)$, respectively. The *attack norm* $p$ chosen under the *attack budget* $\delta$ determines the type of adversarial attack in question, with $p = 2$ and $p = \infty$ as the most common choices. In this paper, we are primarily interested in what we call the *adversarial robustness gap*: $\Delta_p^{\mathrm{adv}} := F_p^{\mathrm{adv}}(\boldsymbol{\theta}; \delta) - F(\boldsymbol{\theta})$, where a small $\Delta_p^{\mathrm{adv}}$ is desirable for adversarial robustness.

**Neural networks**. Our analyses will focus on neural networks under classification. We define a fully connected neural network (FCN) with $\lambda$ hidden layers of $h$ units as below:

$$g(\boldsymbol{x}, \boldsymbol{\theta}) = \mathbf{C}\phi(\mathbf{W}^\lambda \phi(\ldots \mathbf{W}^1 \boldsymbol{x})), \tag{2}$$

where $\boldsymbol{\theta} := (\mathbf{C}, \mathbf{W}^1, \ldots, \mathbf{W}^\lambda)$, $\mathbf{W}^l$ and $\mathbf{C}$ denote hidden layer and linear classification head parameters respectively, and $\phi$ is elementwise ReLU activation function. We omit $\boldsymbol{\theta}$ when it is obvious from the context for brevity. We can write $g$ as the composition of two functions, a linear classifier head $c : \mathbb{R}^h \to \mathbb{R}^{|\mathcal{Y}|}$, and a feature encoder $\Phi : \mathcal{X} \to \mathbb{R}^h$, such that $g(\boldsymbol{x}, \boldsymbol{\theta}) := c(\cdot, \mathbf{C}) \circ \Phi(\cdot, \mathbf{W}^1 \ldots \mathbf{W}^\lambda)(\boldsymbol{x})$. When needed, we use $\boldsymbol{z} = \Phi(\boldsymbol{x})$ or $\boldsymbol{z}_{\mathrm{adv}} = \Phi(\boldsymbol{x}_{\mathrm{adv}})$ to denote latent representations, where $\boldsymbol{x}_{\mathrm{adv}} := \boldsymbol{x} + \boldsymbol{a}^*$. To expedite exposition and reduce notational clutter, throughout our analyses we assume that $\boldsymbol{x} \in \mathbb{R}^h$, and omit bias parameters.

**Lipschitz continuity**. Given two $L^p$ spaces $\mathcal{X}$ and $\mathcal{Y}$, a function $g : \mathcal{X} \to \mathcal{Y}$ is called Lipschitz continuous if there exists a constant $K_p$ such that $\|g(\boldsymbol{x}^1) - g(\boldsymbol{x}^2)\|_p \leq K_p \|\boldsymbol{x}^1 - \boldsymbol{x}^2\|_p, \forall \boldsymbol{x}^1, \boldsymbol{x}^2 \in \mathcal{X}$. Said $K_p$ is called the (global) Lipschitz constant. Any $\bar{K}_p$ that is valid for a subset $\mathcal{U} \subset \mathcal{X}$ is called a local Lipschitz constant on $\mathcal{U}$. Although its computation is NP-hard for even the simplest neural networks (Scaman & Virmaux, 2018); as a notion of input-based volatility, estimation, utilization, and regularization of the Lipschitz constant have been a staple of robustness research (Cisse et al., 2017; Bubeck et al., 2020; Muthukumar & Sulam, 2023; Grishina et al., 2025). Note that the FCN as defined in (2) is Lipschitz continuous in $\ell_p$ for $p \geq 1$, along with other commonly used architectures such as convolutional neural networks (CNN) (Zühlke & Kudenko, 2025).

**Compressibility**. Various prominent approaches to neural network compression exist, such as pruning, quantization, distillation, and conditional computing, (O'Neill, 2020). Here we focus on pruning and low-rank approximation, two of the most commonly used and researched forms of compression (Hohman et al., 2024). More specifically, we focus on inherent properties of network parameters that make them amenable to pruning or low-rank approximation, i.e. their *compressibility*. We will first present a formal definition of a *compressible* vector, and then will show how this definition can be utilized to describe both structured prunability and (approximate) low-rankness.

**Definition 2.1** (($q, k, \epsilon$)-compressibility). *Given a vector $\boldsymbol{\theta} \in \mathbb{R}^d$ and a non-negative integer $k \leq d$, let $\boldsymbol{\theta}_k$ denote the compressed vector which contains the largest (in magnitude) $k$ elements of $\boldsymbol{\theta}$ with all the other elements set to $0$. Then, $\boldsymbol{\theta}$ is ($q, k, \epsilon$)-compressible if and only if*

$$\|\boldsymbol{\theta} - \boldsymbol{\theta}_k\|_q / \|\boldsymbol{\theta}\|_q \leq \epsilon. \tag{3}$$

*In the case of equality, we call $\boldsymbol{\theta}$ strictly ($q, k, \epsilon$)-compressible. Complementarily, the spread variable $\beta \in [0, 1]$ can be used to characterize the dispersion of top-$k$ terms, such that $|\theta_{m_k}| = (1 - \beta)|\theta_{m_1}|$, where $m_i$ indexes the $i$'th largest magnitude element in the vector.*

Moving forward we will assume any vector denoted as compressible is strictly compressible, unless otherwise noted. See the Appendix for a more in-depth discussion of our compressibility definition and how it relates to other notions of approximate sparsity, where we show that our definition distinguishes qualitatively different parameter configurations better compared to prominent alternatives.

**Structured compressibility**. Importantly, given that the $\boldsymbol{\theta}$ can be any vector, the above definition can be used flexibly to describe different notions of compressibility, including those of structured compressibility, where particular substructures in the model dominate the rest. More specifically, given a layer parameter matrix $\mathbf{W} \in \mathbb{R}^{h \times h}$ from (2), let $\boldsymbol{\nu} := (\|\mathbf{w}_1\|_1, \ldots, \|\mathbf{w}_h\|_1)$ denote $\ell_1$ norms of rows of the matrix $\mathbf{W}$. The compressibility of $\boldsymbol{\nu}$ would correspond to *row/neuron compressibility*, which is a desirable property for neural network parameters as it expedites pruning of whole neurons, with tangible computational gains. Note that this also would correspond to filter compressibility/prunability in CNNs with a matricization of the convolution tensor. Similarly, let $\boldsymbol{\sigma} := (\sigma_1, \sigma_2, \ldots)$ denote the singular values of matrix $\mathbf{W}$. Compressibility of $\boldsymbol{\sigma}$ would correspond to *spectral compressibility*, serving as a notion of approximate/numerical low-rankness.

## 3 NORM-BASED ADVERSARIAL ROBUSTNESS BOUNDS

**Motivating hypothesis**. Although structured (neuron, spectral) compressibility is desirable from a computational perspective, it also focuses the total energy of the parameters on a few dominant terms (rows/filters, singular values). This in turn creates a few potent directions in the latent space and increases the operator norms of the parameters ($\ell_\infty$, $\ell_2$ operator norms respectively).

This increases their sensitivity to worst-case perturbations: adversarial attacks exploiting these directions are amplified in the representation space, and can more easily disrupt the predictions of the model. For a more specific example using spectral compressibility, given a single layer neural network $g(\boldsymbol{x}) = \mathbf{C}\phi(\mathbf{W}\boldsymbol{x})$, assume that $\sigma_1 \gg \sigma_{j\neq 1}$, i.e. first singular value dominates the rest under high compressibility. Then, an adversarial perturbation $\boldsymbol{a}$ that aligns with the associated right singular vector $\boldsymbol{v}_1$ s.t. $\boldsymbol{v}_1^T \boldsymbol{a}/\|\boldsymbol{a}\|_2 \approx 1$, will have scaled their post-layer representation by approximately $\sigma_1$. This in turn would facilitate them to dominate the latent space against the original image, and ultimately change the prediction of the model. Taken from an experiment presented in full detail in Section 4, Figure 2 visualizes this phenomenon in reality. Here, we utilize PCA to visualize the input image, adversarial perturbation, and decision boundaries for a single

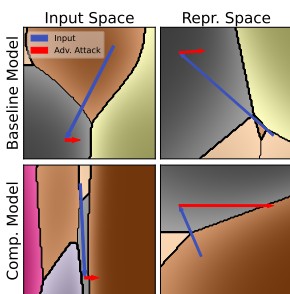

Figure 2: Decision boundaries under compressibility.

sample under a baseline vs. compressible (low-rank) model. The top row visualizes the baseline model, where the minuscule adversarial perturbation fails to move the perturbed image across class boundaries. The bottom row however, illustrates the compressible model under attack. Here, although attack budget is identical in the input space, the adversarial perturbation is dramatically amplified in the representation space, leading to a successful adversarial attack. Note that the decision boundaries in compressible model's input space is much more contracted to reflect this vulnerability. In the Appendix, we dedicate a section to providing a stronger, step-by-step intuition for our hypotheses.

**Compressibility-based Lipschitz bounds.** Our theory will relate structured compressibility to robustness through its effect on the network's operator norms and Lipschitz constants. However, this brings about a particular conceptual challenge. Our notion of $(q, k, \epsilon)$-compressibility, like others' (Diao et al., 2023), is a *scale-independent* measure. Therefore, any direct relation between compressibility and Lipschitz constants would be rendered void by the arbitrary scaling of the parameters. Therefore, we characterize $\ell_\infty$ and $\ell_2$ operator norms of the parameters by an upper bound that decomposes into (compressibility × Frobenius norm) terms. This "structure vs. scale" decomposition allows us to meaningfully relate compressibility and robustness, and also allows us to develop concrete hypotheses regarding the effect of various interventions in neural network training.

**Theorem 3.1.** *The following statements relate operator norms and structured compressibility.*

*(a) Neuron compressibility (i.e. row-sparsity):* Let $\mathbf{w}_i, i \in [h]$ denote the rows of the matrix $\mathbf{W}$, and let $\boldsymbol{\nu} := (\|\mathbf{w}_1\|_1, \ldots, \|\mathbf{w}_h\|_1)$ denote $\ell_1$ norms of its rows. Assuming $\boldsymbol{\nu}$ is $(1, k_{\boldsymbol{\nu}}, \epsilon_{\boldsymbol{\nu}})$ compressible and each row $\mathbf{w}_i$ is $(2, k_r, \epsilon_r)$-compressible implies:

$$\|\mathbf{W}\|_\infty \leq \frac{(1 - \epsilon_{\boldsymbol{\nu}})}{(1 - \beta_{\boldsymbol{\nu}})} \left( \frac{\sqrt{hk_r} + h\epsilon_r}{k_{\boldsymbol{\nu}}} \right) \|\mathbf{W}\|_F. \qquad (4)$$

*(b) Spectral compressibility (i.e. low-rankness):* Let $\boldsymbol{\sigma} := (\sigma_1, \sigma_2, \ldots)$ denote the singular values of matrix $\mathbf{W}$. Assuming $\boldsymbol{\sigma}$ is $(1, k_{\boldsymbol{\sigma}}, \epsilon_{\boldsymbol{\sigma}})$-compressible implies:

$$\|\mathbf{W}\|_2 \leq \frac{(1 - \epsilon_{\boldsymbol{\sigma}})}{(1 - \beta_{\boldsymbol{\sigma}})} \left( \frac{\sqrt{h}}{k_{\boldsymbol{\sigma}}} \right) \|\mathbf{W}\|_F. \qquad (5)$$

Intuitively, Theorem 3.1 describes how increasing compressibility affects layer operator norms: Neuron compressibility, *i.e.* a small number of rows dominating the matrix increases $\ell_\infty$ operator norm of the matrix, especially if the spread within these dominant rows are high. Similarly, increased spectral compressibility and spread increases the $\ell_2$ operator norm. Note that the latter result is closely related to results from the literature that connect stable rank or condition number to robustness (Savostianova et al., 2023; Feng et al., 2025), see Section 5. Although Theorem 3.1 directly relates neuron and spectral compressibility to perturbations defined in $\ell_\infty$ and $\ell_2$ norms, standard norm inequalities couple these operator norms up to dimension factors, so vulnerability trends transfer

across $\ell_\infty$ and $\ell_2$ settings (see Appendix for further discussion and results). Lastly, while we utilize the upper bounds for our following theoretical results, additional theoretical results in the Appendix characterize lower bounds on the operator norm with similar implications.

As we move on to characterizing layers within a neural network, $\mathbf{W}_k^l$ will be used to denote the *compressed* version of the parameter matrix of layer $l$. In the case of row compression, this corresponds to setting the $h - k$ trailing rows to $\mathbf{0}$. In the case of spectral compression, given the singular value decomposition (SVD), $\mathbf{W}^l = \mathbf{U}^l \mathbf{\Sigma}^l \mathbf{V}^{l^T}$, the compressed matrix corresponds to $\mathbf{W}_k^l := \mathbf{U}_k^l \mathbf{\Sigma}_k^l \mathbf{V}_k^{l^T}$, where the $h - k$ smallest singular values are truncated.

Note that the sensitivity of the network not only relies on the characteristics of layer parameters, but also on the interactions between them. For example, it is possible to upper bound the operator norm of two consecutive layers interleaved by a ReLU nonlinearity with $\|\mathbf{W}^{l+1}\| \|\mathbf{W}^l\|$. However, this is an overly pessimistic bound, as it accounts for the most potent directions of each layer perfectly lining up (unlikely in reality), and ignores the nonlinearity. This is why for our following theorem, we first introduce the interlayer alignment terms $A_p$: These terms will help improve the operator norm bound by correcting for the said overly pessimistic assumption by using the "alignment" of the top-$k$ terms in each layer - see Scaman & Virmaux (2018) for a similar approach. With $\mathcal{D}$ as the set of all diagonal binary matrices (for ReLU activations), we define $A_p$, for $p \in \{2, \infty\}$ as:

$$A_\infty(l) \triangleq \max_{\mathbf{D} \in \mathcal{D}} \frac{\|\mathbf{W}_k^{l+1} \mathbf{D} \mathbf{W}_k^l\|_\infty}{\|\mathbf{W}^{l+1}\|_\infty \|\mathbf{W}^l\|_\infty} + R_\infty, \quad A_2(l) \triangleq \max_{\mathbf{D} \in \mathcal{D}} \frac{\|\sqrt{\Sigma_k^{l+1}} \mathbf{V}_k^{l+1^T} \mathbf{D} \mathbf{U}_k^l \sqrt{\Sigma_k^l}\|_2}{\sqrt{\|\mathbf{W}^{l+1}\|_2 \|\mathbf{W}^l\|_2}} + R_2, \tag{6}$$

where $R_\infty, R_2$ are remainder alignment terms defined and shown to be $R_p \to 0$ as $\epsilon \to 0$ in the Appendix. We refer the reader to our proofs in the Appendix to explain the exact form the alignment terms take and a comparison to previous approaches, where we also dedicate a section to provide a more intuitive understanding for them. Having Theorem 3.1 to help characterize the compressibility-based sensitivity of layers, and (6) to help connect them, we now provide an upper bound to the Lipschitz constant of the complete encoder network.

**Theorem 3.2.** *Let $L_\Phi^p$ be the Lipschitz constant of the encoder $\Phi$ defined following* (2). *Let $\mathcal{D}$ denote the set of all diagonal binary matrices, corresponding to ReLU activation layers. Then:*

*(a) Neuron compressibility: The $\ell_\infty$ Lipschitz constant of $\Phi$ can be upper bounded by:*

$$L_\Phi^\infty \leq \hat{L}_\Phi^\infty := \prod_{l=1}^\lambda \frac{(1 - \epsilon_{\boldsymbol{\nu}})}{(1 - \beta_{\boldsymbol{\nu}})} \left( \frac{\sqrt{hk_r} + h\epsilon_r}{k_{\boldsymbol{\nu}}} \right) \|\mathbf{W}^l\|_F \prod_{l=1}^{\lambda-1} \tilde{A}_\infty(l), \tag{7}$$

*where $\tilde{A}_\infty(l) = A_\infty(l)$ if $l \in S_{opt}$, and 1 otherwise. $S_{opt} \subseteq \{1, 2, \ldots, \lambda - 1\}$ is the optimal alignment partition set (See Definition A.4) that can be determined in $O(\lambda)$ time.*

*(b) Spectral compressibility: The $\ell_2$ Lipschitz constant of $\Phi$ can be upper bounded by:*

$$L_\Phi^2 \leq \hat{L}_\Phi^2 := \prod_{l=1}^\lambda \frac{(1 - \epsilon_{\boldsymbol{\sigma}})}{(1 - \beta_{\boldsymbol{\sigma}})} \left( \frac{\sqrt{h}}{k_{\boldsymbol{\sigma}}} \right) \|\mathbf{W}^l\|_F \prod_{l=1}^{\lambda-1} A_2(l). \tag{8}$$

Note that for brevity and without loss of generality we assume uniform compressibility across layers. These upper bounds can be directly used in conjunction with other results from the literature (Ribeiro et al., 2023) to characterize adversarial robustness gap, as demonstrated in the next corollary. Here, given the binary classification context, we assume $\mathbf{C} \in \mathbb{R}^h$ and $\hat{y} \in \mathbb{R}$ with slight abuse of notation.

**Corollary 3.3.** *Under a binary classification task with logistic loss, $\ell(y, \hat{y}) = \log\left(1 + e^{-y\hat{y}}\right)$, given a neural network classifier as described in* (2)*, under the same assumptions with* (7) *and* (8)*, we have $F_\infty^{adv}(\boldsymbol{\theta}; \delta) \leq F(\boldsymbol{\theta}) + \delta \hat{L}_\Phi^\infty \|\mathbf{C}\|_1$ and $F_2^{adv}(\boldsymbol{\theta}; \delta) \leq F(\boldsymbol{\theta}) + \delta \hat{L}_\Phi^2 \|\mathbf{C}\|_2$, respectively.*

See also results by Nern et al. (2023) that connect encoder sensitivity to adversarial robustness. Note that although bounds provided in Theorem 3.2 are tighter than the pessimistic "product-of-norms" bounds, they deliberately *trade off* some tightness by utilizing Theorem 3.1. However, in return, this results in bounds that decomposes into analytically interpretable and actionable terms. Such bounds have proven valuable in analyzing adversarial robustness in deep learning (Wen et al.,

2020). Regardless, Figure 3 demonstrates the close correlation our bounds shows with the empirical robustness gap ($\rho = 0.947$), in a 2-hidden-layer neural network with varying spectral compressibility (obtained through systematically varying the rank of factorized layer matrices). We provide full details in the Appendix, where we also explore the alignment terms' empirical behavior and estimation techniques, although a detailed analysis thereof lies beyond our primary focus.

Given our focus on compressibility-driven threats to structural encoder safety under potential distribution shifts, we upper bound encoder's global Lipschitz constant. While approaches that utilize local Lipschitzness are known to produce tighter bounds (Cisse et al., 2017; Roth et al., 2020), results in Figure 3 and Section 4 show that for our theory this does not come at the cost of predictive power. Our results with universal adversarial examples, to be presented within the latter, particularly highlight the relevance of global structural vulnerabilities. Lastly, in the Appendix we demonstrate strong correlation between global Lipschitz upper bounds and empirically estimated local Lipschitz constants.

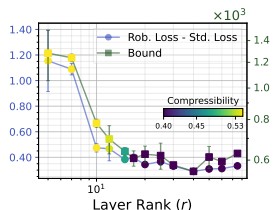

Figure 3: Corollary 3.3 vs. empirical robustness gap.

## 4 EXPERIMENTS

We now validate our theoretical findings through systematic experimentation. We first validate our *motivating hypothesis* and then empirically show that (i) neuron and spectral compressibility-inducing interventions will reduce adversarial robustness against $\ell_\infty$ and $\ell_2$ adversarial attacks; (ii) the negative effects of compressibility to persist under adversarial training, (iii) the compressibility-related vulnerabilities induced on representations during pretraining will impact any downstream task in transfer learning; (iv) increasing compressibility creates vulnerable directions in the latent space, further enabling universal adversarial examples (UAEs), while increasing Frobenius norm will create vulnerability without leading to UAEs; and (v) compressed models will inherit the vulnerability of the original models, and conducting compression based on $(q, k, \epsilon)$-compressibility, reducing the spread of the dominant terms, or regularizing interlayer alignment will improve robustness.

**Datasets, architectures, and training**. We conduct our experiments in the most commonly used datasets and architectures in the literature on adversarial robustness and compression (Piras et al., 2025). Datasets we use include MNIST (Deng, 2012), CIFAR-10, CIFAR-100 (Krizhevsky & Hinton, 2009), SVHN (Netzer et al., 2011), Flickr30k (Young et al., 2014), and ImageNet-1k (Deng et al., 2009). Architectures we utilize include fully connected networks (FCN), ResNet18 (He et al., 2016), VGG16 (Simonyan & Zisserman, 2014), WideResNet-101-2 (Zagoruyko & Komodakis, 2016), vision transformer (ViT) - both as a standalone classifier (Dosovitskiy et al., 2021) and as part of a CLIP encoder (Radford et al., 2021), and Swin Transformer (Liu et al., 2021). Unless otherwise noted, we use softmax cross-entropy loss, the AdamW optimizer with a weight decay of $0.01$, a learning rate of $0.001$, and use a validation set based model selection for early stopping.

**Evaluating and training for adversarial robustness**. When evaluating adversarial robustness, we utilize AutoPGD as the primary adversarial attack algorithm for evaluation (Croce & Hein, 2020), as implemented by Nicolae et al. (2018). When training for adversarial robustness, we utilize a PGD attack to generate adversarial samples at every iteration (Madry et al., 2018). Unless otherwise noted, we use a ratio of 0.5 for adversarial samples in a training minibatch. We use $\delta = 8/255$ and $\delta = 0.5$ for $\ell_\infty$ and $\ell_2$ attacks respectively for end-to-end adversarially trained models. We use $0.25\times$ of these budgets for evaluating standard trained or adversarially fine-tuned models to allow a visible comparison (See Appendix for qualitatively identical results under different budgets and attack algorithms). By default, we present results for $\ell_\infty$ and $\ell_2$ attacks when evaluating robustness under neuron and spectral compressibility respectively, and defer the cross-norm results to the supplementary material, which also includes further details on our experiment settings and implementation.

**Comparison across methods**. Given that our theory is agnostic to the source of structured compressibility, we experiment and confirm our predictions with various methods to induce compressibility. Therefore, to retain the equivalence between these different methods and prevent confounding from specific compression procedures, we primarily compare uncompressed (e.g. unpruned) models while explicitly highlighting their different levels of compressibility. While some approaches such as low-rank factorization do not involve a separate compression step, in approaches where a specific compression procedure is commonly utilized in practice (e.g. filter pruning after regularized training), we show that our results apply to the compressed models as well.

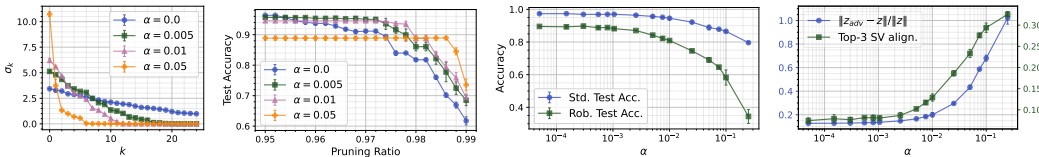

Figure 4: Model statistics under increasing strength of nuclear norm regularization ($\alpha$).

## 4.1 RESULTS

**Testing the motivating hypothesis.** We start our empirical analysis with a demonstrative experiment to visually investigate the implications of our motivating hypothesis. For this, we train a single 400-width hidden layer FCN with ReLU activations on the MNIST dataset. We use nuclear norm regularization (NNR) to encourage spectral compressibility, adding the term $\alpha\|\boldsymbol{\sigma}\|_1$ to the training objective, with $\alpha$ as a hyperparameter. To avoid confounding by NNR decreasing overall parameter norms, we apply Frobenius norm normalization to $\mathbf{W}^1$ at every iteration (Miyato et al., 2018). While our following experiments will utilize more practically relevant norm control mechanisms, we currently apply normalization to fully isolate the effects of compressibility.

In Figure 4 (left) we validate that our intervention indeed increases spectral norm compressibility. As expected, Figure 4 (center left) shows that spectral compressibility actually allows pruning: the more compressible models retain their performance under stronger spectral pruning. Figure 4 (center right) shows that increased compressibility comes at the cost of adversarial robustness: as $\alpha$ increases, adversarial accuracy dramatically falls. We further investigate whether this fall is due to our hypothesized mechanism. We let $\boldsymbol{z} = \Phi(\boldsymbol{x})$ and $\boldsymbol{z}_{\text{adv}} = \Phi(\boldsymbol{x} + \boldsymbol{a}^*)$ denote the learned representations of clean and perturbed input images. If the adversarial attacks are taking advantage of the

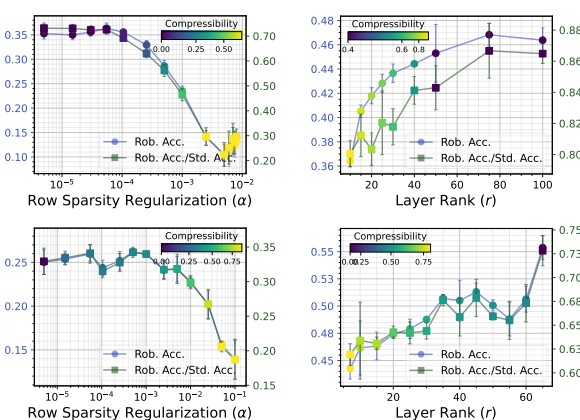

Figure 5: Results with FCN (top) and ResNet18 (bottom) trained on CIFAR-10 dataset.

potent directions created by compressibility, then as compressibility increases: (1) The perturbations $\boldsymbol{a}^*$ should align more with the dominant singular directions, *i.e.*, $\mathbf{v}_i^{\mathsf{T}}\boldsymbol{a}^* \gg \mathbf{v}_j^{\mathsf{T}}\boldsymbol{a}^* \;\forall i \in [k], j \notin [k]$, (2) representations of adversarial perturbations should grow stronger in relation to the original image's representation, *i.e.* $\|\boldsymbol{z}_{\text{adv}} - \boldsymbol{z}\|_2/\|\boldsymbol{z}\|_2$ should increase. Results presented in Figure 4 (right) confirm both predictions, further supporting our motivating hypothesis. Lastly, the previously presented Figure 2 visualizes the effect of compressibility in the input and representation space. We provide a more detailed, step-by-step account of how potent leading directions are exploited by white box and black box adversaries in the Appendix for stronger intuition.

**Adversarial robustness and compressibility under standard training.** For implications of our analysis under more realistic settings, we start by investigating the effects of compressibility on adversarial robustness in fully connected networks (FCN). We induce neuron and spectral compressibility through group lasso regularization[1] and low-rank factorization, respectively (latter avoids the excessive cost of nuclear norm regularization). As above, we conduct Frobenius norm normalization at every iteration. Figure 5 (top) presents the results of these experiments: The reduction in adversarial robustness as a function of increasing compressibility is clear in both cases, confirming our main hypothesis. Note that we present robust accuracy (RA) / standard accuracy (SA) ratio alongside RA to highlight that the obtained results are not due to baseline SA being lower under compressibility.

We then investigate whether our hypotheses apply beyond the context of our theory, starting with convolutional neural networks (CNNs). We first test our predictions in ResNet18 models trained on CIFAR-10 datasets. Here we eschew Frobenius norm normalization for standard weight decay.

---

[1]Group lasso regularization penalizes the $\ell_1$ norm of row $\ell_2$ norms of each layer, promoting row-sparsity.

However, to prevent confounding from group lasso's effect on parameter scales, we create a scale-invariant version that regularizes row norms' $\ell_1/\ell_2$ norm ratio.[2] Figure 5 (bottom) demonstrates that the above effects clearly translate to this setting as well, further solidifying the relationship between structured compressibility and adversarial robustness. We present

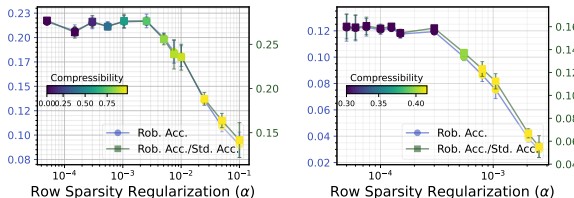

Figure 6: Results with ViT (left) and CLIP (right).

similar results on two other architectures (VGG16, WideResNet-101) and two other datasets (CIFAR-100, SVHN) in the Appendix. Going forward, for brevity we will focus on neuron compressibility results, and defer corresponding spectral compressibility results to the Appendix, where we also discuss unstructured compressibility and inductive bias-based emergent compressibility.

**Experiments with transformers.** We next test our hypotheses under transformer architectures. Figure 6 (left) replicates our results under a ViT classifier model trained on CIFAR-10 dataset. Further, to test whether our hypothesis holds under a zero-shot classification setting, we fine-tune a pre-trained CLIP model on Flickr30k dataset under varying degrees of sparsification regularization, and conduct standard and adversarial zero-shot classification using ImageNet-1k dataset. We find that our results (Figure 6, right) replicate here as well. That simply fine-tuning with sparsification can create this vulnerability with commonly repurposed encoder backbones highlights the safety implications of our results. See Appendix for further details and findings under other training settings.

**Effects of compressibility on robustness under adversarial training.** Given that adversarial training is the primary method for obtaining models that are robust against adversaries, we next investigate whether the effects we have observed will persist under this regime. To make this setting as close to practice as possible, we also include a learning rate annealing schedule (Cosine annealing) and basic data augmentation (random horizontal flip and crops), as well as attacks with standard budgets as described above. The results almost identically replicate our observations under standard training (Figure 7, left). Although adversarial training increases adversarial robustness overall, the relative effect of compressibility remains as it is.

**Universal adversarial examples.** Examining the terms in Theorem 3.2, we predict that while both compressibility and Frobenius norm are likely to increase vulnerability, only the former is likely to lead to universal adversarial examples (UAEs) (Moosavi-Dezfooli et al., 2017), due to the global vulnerable directions it creates. To test our hypothesis, we modify the setting of FCN experiments presented above: In contrast to increasing row sparsity regularization under a fixed Frobenius norm, in an alternative set of experiments we systematically increase the constant to which Frobenius norm of the layers is fixed, without any row sparsity regularization. We utilize a FGSM-based (Goodfellow et al., 2015) UAE computation to develop adversarial samples. Figure 7 (center left, center right) confirms our hypothesis: while increasing Frobenius norm only decreases standard adversarial robustness, increasing compressibility *additionally* creates vulnerability to UAEs. In the Appendix, we replicate these results under a ResNet18. Importantly, we also show that the converse relationship also holds: Training against UAEs vs. standard adversarial samples decreases top-$k$ parameter spread $\beta$, providing further support for our arguments.

**Adversarial vulnerability under transfer learning.** Next, we investigate our hypothesis that the effects of compressibility should persist under transfer learning due to the structural effects created on representations. We train a ResNet18 model on CIFAR-100 dataset with increasing row sparsity regularization. After the training is complete, we freeze the encoder parameters and train a linear classifier head for prediction on CIFAR-10 dataset and evaluate the robustness of the resulting model. Figure 7 (right) shows that the effects of compressibility observed above directly translate to the context of transfer learning, where increased compressibility in pretraining affects robustness performance in the downstream task, for which the network is fine-tuned.

**Pruning and robustness.** While we extensively investigated the effects of compressibility on robustness, for neuron compressibility we now focus on the behavior of models under downstream layerwise filter pruning to ensure our insights transfer to practical pruning scenarios. Us-

---

[2]In the Appendix, we show that standard group lasso creates a tug-of-war between increasing compressibility and decreasing parameter scales; the former eventually wins, resulting in decreased robustness.

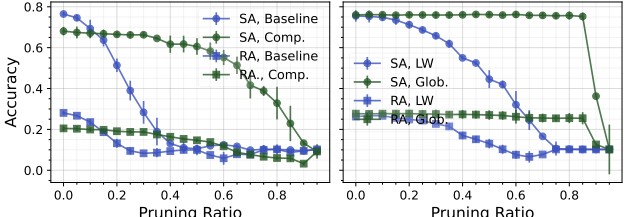

Figure 7: (Left) Effects of compressibility under adversarial training. UAEs under increasing (center left) compressibility vs. (center right) parameter scale. (Right) Robustness under transfer learning.

ing the ResNet18 and CIFAR-10 combination under adversarial training, in Figure 8 (left), we compare the baseline model ($\alpha = 0.0$) to a model regularized to be compressible ($\alpha = 0.1$).

We see that at no point do the compressed models surpass the uncompressed performance of the baseline model in terms of standard and robust accuracy. However, as pruning ratio increases, the baseline model fails to retain its standard and robust performance, whereas the compressible (sparsified) model does considerably better, demonstrating the fundamental tension between robustness and compressibility. In the Appendix, we show that these results hold after post-pruning fine-tuning as well. Additionally, there we demonstrate that post-pruning fine-tuning can act as an additional source of vulnerability in and of itself, as under this procedure adversarial robustness deteriorates much faster than standard accuracy, confirming our results under yet another source of norm imbalance.

Figure 8: Robustness under compression. SA/RA: Standard/Robust Acc. LW/Glob.: Layerwise vs. global pruning.

In Figure 8 (right), we show that conducting pruning based on two simple interventions inspired by our bounds results in tangible improvements in standard and robust performance under pruning. Given the fact that layerwise pruning is known to produce harmful bottlenecks that lead to layer collapse (Blalock et al., 2020), instead of targeting a pruning ratio and pruning each layer accordingly, we set a target $\epsilon$ for each layer, and for each compute $k$ that satisfies this $\epsilon$ level. Given a target global pruning ratio, we scan over different levels of $\epsilon$ and determine the level that gets closest to the target ratio. Moreover, during training we control the spread of the dominant terms, $\beta$, which our analyses show to be harmful for robustness, without decreasing compressibility. We accomplish this through regularizing the variance of the top $0.05$ of each layer's filters' norms. Figure 8 (right) demonstrates that our interventions create a tangible improvement in performance retention. In the Appendix, we provide additional results showing that interlayer alignment can also be successfully used as a regularization target for robust compressibility. We consider these interventions both as validations of our theory and promising directions for future robust compression research. However, we also highlight that it may not be possible to completely negate the dangers of concentrating parameter energy in few substructures, extensively demonstrated by our theory and experiments. Therefore, while pruning and low-rank approximation remain valuable compression methods, combining intermediate levels thereof with other compression methods such as quantization or knowledge distillation seems to be the most promising approach in reconciling safety and robustness, which is in line with recent findings in the literature (Pavlitska et al., 2023).

## 5 RELATED WORK

**Adversarial robustness**. The susceptibility of the neural network models to adversarial examples created through small perturbations (Szegedy et al., 2014) engendered a lot of research investigating the issue (Madry et al., 2018). To this day adversarial robustness remains one of the most important topics in machine learning safety (Malik et al., 2024). The literature ranges from the development of new attacks and defenses (Moosavi-Dezfooli et al., 2016; Abdollahpoorrostam et al., 2024), to investigating sources/mechanisms of adversarial vulnerability, to implications of AEs for the inductive biases of modern machine learning architectures (Ilyas et al., 2019; Ortiz-Jimenez et al., 2021; Xu et al., 2024), to developing strategies to retain model expressivity and generalization while defending against adversarial attacks (Tsipras et al., 2019; Zhang et al., 2024).

**Pruning and low-rank approximation**. Prominent compression approaches include pruning, quantization, distillation, conditional computing, and efficient architecture development (O'Neill, 2020). Out of these, pruning remains among the most actively researched compression approaches due to its versatility (Cheng et al., 2024). Inducing compressibility / sparsity at training time is one of the easiest way to obtain prunable models (Hohman et al., 2024). Compressibility across different substructures, a.k.a. group sparsity (Li et al., 2020b), allows for structured pruning (e.g. neuron/row, filter/channel, kernel pruning), which is computationally efficient (Yang et al., 2018), yet leads to a sharp reduction in network connectivity, threatening performance (Blalock et al., 2020). Lastly, spectral compressibility relaxes the notion of low-rankness (Suzuki et al., 2020; Schotthöfer et al., 2022). While nuclear norm regularization is not a commonly utilized intervention due to the computational costs involved, low-rank factorization continues to be a prominent architectural design choice due to its attractive theoretical and empirical properties (Savostianova et al., 2023).

**Compressibility and robustness**. Whereas some research argues that compressibility/sparsity is beneficial for adversarial robustness (Guo et al., 2018; Balda et al., 2020; Liao et al., 2022), others indicate the relation is *at best* highly dependent on the degree and type of compressibility, as well as attack type (Li et al., 2020a; Merkle et al., 2022; Savostianova et al., 2023; Feng et al., 2025). While a stream of new methods incorporate adversarial robustness in novel ways to pruning (a.k.a. *adversarial pruning*), recent systematic benchmarks reveal marginal benefits for such methods compared to weight-based pruning (Lee et al., 2020; Piras et al., 2025). Whereas some methods demonstrate benefits of adversarial training-aware sparsification (Gui et al., 2019; Sehwag et al., 2020; Pavlitska et al., 2023), adversarial training hampers standard generalization, transferability as well as computational feasibility especially for larger models, plaguing such methods (Tsipras et al., 2019; Wen et al., 2020; Yang et al., 2024). A comprehensive understanding of how compressibility and robustness interact, adversarial or otherwise (Barsbey et al., 2025), is still lacking.

**Comparison to previous research**. Our work addresses a critical gap in the literature: paucity of research that establishes a principled, theoretical relationship between structured compressibility and adversarial robustness with extensive empirical confirmation. While doing so, we find that it produces complementary results to most closely related previous work. For example, Savostianova et al. (2023) and Feng et al. (2025) highlight the adversarial vulnerability created by increased condition numbers due to high unstructured sparsity or low-rank training, respectively. Our results complement and extend their conclusions by providing convergent theoretical results with a more fine-grained, source-agnostic notion of compressibility, and can naturally incorporate neuron compressibility/prunability, which the cited work do not address. Lastly, in our Appendix we investigate two prominent structured adversarial pruning methods (Zhao & Wressnegger, 2023; Zhong et al., 2023; Piras et al., 2025), and demonstrate that these *implicitly* control operator norms in a way that cannot be simply attributed to adversarial training. Our complementary findings highlight the design of theoretically informed robust pruning methods as a promising future research direction.

## 6    Conclusion and future work

In this paper, we present a unified theoretical and empirical treatment of how structured compressibility shapes adversarial robustness. Via a novel analysis of neuron-level and spectral compressibility, we uncover a fundamental mechanism: compression concentrates sensitivity along a small number of directions in representation space, rendering models more vulnerable - even under adversarial training and transfer learning. Our norm-based robustness bounds offer interpretable decompositions that predict both standard and universal adversarial vulnerability, and shed light on the trade-offs between efficiency and safety in modern neural networks. Empirically, we validate these insights across datasets, architectures, and training regimes, showing how compressibility determines adversarial susceptibility in various learning contexts. Inspired by our bounds, we outline simple, targeted strategies that can mitigate these vulnerabilities.

**Future work**. While our theory provides novel insights into structured compressibility - adversarial vulnerability relationships, future work must focus on composite effects of practical compression approaches. A structured compression scheme might have multiple effects simultaneously: while harming robustness through increased structural imbalance, it can help it by reducing Frobenius norms or interlayer alignment, or closing off-data-manifold directions in the representation space. Achieving tighter local bounds, and incorporating other types of compression (e.g. semi-structured pruning, quantization) and distribution shifts (e.g. other $\ell_p$ attacks, spurious correlations) are other important future directions.

ACKNOWLEDGEMENTS

MB was supported by the EPSRC Project GNOMON [EP/X011364/1]. UŞ was partially supported by the French government under the management of Agence Nationale de la Recherche as part of the "Investissements d'avenir" program, reference ANR-19-P3IA-0001 (PRAIRIE 3IA Institute) and by the European Research Council Starting Grant DYNASTY – 101039676. TB was supported by a UKRI Future Leaders Fellowship [grant number MR/Y018818/1].

REPRODUCIBILITY STATEMENT

Experiment details are provided in the main paper and in the supplementary material. Repository at `https://github.com/mbarsbey/advcomp` includes code for reproducing main results.

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

# On the Interaction of
# Compressibility and Adversarial Robustness
# –Appendix–

## Contents

# A  PROOFS

We start with a number of auxiliary results that are used in the theorems and corollary presented in Section 3.

**Lemma A.1.** *For any strictly $(q, k, \epsilon)$compressible vector $\boldsymbol{\theta}$ and for all $q \geq 1$, $\|\boldsymbol{\theta}^{(k)}\|_q = (1 - \epsilon^q)^{1/q}\|\boldsymbol{\theta}\|_q$.*

*Proof.* $\|\boldsymbol{\theta} - \boldsymbol{\theta}^{(k)}\|_q^q = \epsilon^q\|\boldsymbol{\theta}\|_q^q$ follows from the definition of compressibility. Adding $\|\boldsymbol{\theta}^{(k)}\|_q^q$ to both sides leads to $\|\boldsymbol{\theta}\|_q^q = \epsilon^q\|\boldsymbol{\theta}\|_q^q + \|\boldsymbol{\theta}^{(k)}\|_q^q$, with LHS due to elements of $\boldsymbol{x}$ and $\boldsymbol{\theta} - \boldsymbol{\theta}^{(k)}$ populating disjoint sets of coordinates. Result follows with simple algebraic manipulation. □

Note that for the results in this section, we use $\boldsymbol{\theta}^{(k)}$ and $\boldsymbol{\theta}_k$ equivalently to denote a vector that includes only the $k$ dominant terms.

**Lemma A.2.** *For $p^* < q$, given the $(2, k, \epsilon)$-compressible vector $\boldsymbol{\theta} \in \mathbb{R}^d$, we have:*

$$\|\boldsymbol{\theta}\|_{p^*} \leq k^{\frac{1}{p^*} - \frac{1}{q}}\|\boldsymbol{\theta}^{(k)}\|_q + d^{\frac{1}{p^*} - \frac{1}{q}}\epsilon\|\boldsymbol{\theta}\|_q. \tag{9}$$

*Proof.* We start by applying Minkowski's inequality to $\|\boldsymbol{\theta}\|_{p^*}$:

$$\|\boldsymbol{\theta}\|_{p^*} \leq \|\boldsymbol{\theta}^{(k)}\|_{p^*} + \|\boldsymbol{\theta} - \boldsymbol{\theta}^{(k)}\|_{p^*}. \tag{10}$$

We now bound the terms on RHS separately. For the first term, since $p^* < q$ by Hölder's inequality for k-sparse vectors we have

$$\|\boldsymbol{\theta}^{(k)}\|_{p^*} \leq k^{\frac{1}{p^*} - \frac{1}{q}}\|\boldsymbol{\theta}^{(k)}\|_q.$$

For the next term, we can write

$$\|\boldsymbol{\theta} - \boldsymbol{\theta}^{(k)}\|_{p^*} \leq d^{\frac{1}{p^*} - \frac{1}{q}}\|\boldsymbol{\theta} - \boldsymbol{\theta}^{(k)}\|_q \leq d^{\frac{1}{p^*} - \frac{1}{q}}\epsilon\|\boldsymbol{\theta}\|_q,$$

with the left inequality due to Hölder's inequality, and the right due to $\boldsymbol{\theta}^{(k)}$'s $(2, k, \epsilon)$-compressibility. Combining the expressions for both terms, we have

$$\|\boldsymbol{\theta}\|_{p^*} \leq k^{\frac{1}{p^*} - \frac{1}{q}}\|\boldsymbol{\theta}^{(k)}\|_q + d^{\frac{1}{p^*} - \frac{1}{q}}\epsilon\|\boldsymbol{\theta}\|_q. \tag{11}$$

□

**Proposition A.3.** *Given a linear binary classifier and binary cross-entropy loss function, assuming $\boldsymbol{\theta} \in \mathbb{R}^h$, we have the following bound:*

$$F_p^{\mathrm{adv}}(\boldsymbol{\theta}; \delta) \leq F(\boldsymbol{\theta}) + \delta\|\boldsymbol{\theta}\|_{p^*} \tag{12}$$

*Proof of Proposition A.3.* For binary cross-entropy loss we have:

$$f^{\mathrm{adv}}(\boldsymbol{x}, \boldsymbol{\theta}; \delta) = \log\left(1 + \exp\left(-y(\boldsymbol{x}^\top\boldsymbol{\theta}) + \delta\|\boldsymbol{\theta}\|_{p^*}\right)\right).$$

We observe that $f^{\mathrm{adv}}(\boldsymbol{x}, \boldsymbol{\theta}; \delta) \leq f(\boldsymbol{x}, \boldsymbol{\theta}; \delta) + \delta\|\boldsymbol{\theta}\|_{p^*}$ since

$$f^{\mathrm{adv}}(\boldsymbol{x}, \boldsymbol{\theta}; \delta) = \log\left(1 + \exp\left(-y(\boldsymbol{x}^\top\boldsymbol{\theta}) + \delta\|\boldsymbol{\theta}\|_{p^*}\right)\right)$$

$$= \log\left(1 + \exp\left(-y(\boldsymbol{x}^\top\boldsymbol{\theta})\right)\right) + \log\left(\frac{1 + \exp\left(-y(\boldsymbol{x}^\top\boldsymbol{\theta}) + \delta\|\boldsymbol{\theta}\|_{p^*}\right)}{1 + \exp\left(-y(\boldsymbol{x}^\top\boldsymbol{\theta})\right)}\right)$$

$$= f(\boldsymbol{x}, \boldsymbol{\theta}; \delta) + \log\left(1 + (\exp\left(\delta\|\boldsymbol{\theta}\|_{p^*}\right) - 1)\frac{\exp\left(-y(\boldsymbol{x}^\top\boldsymbol{\theta})\right)}{1 + \exp\left(-y(\boldsymbol{x}^\top\boldsymbol{\theta})\right)}\right)$$

$$\leq f(\boldsymbol{x}, \boldsymbol{\theta}; \delta) + \delta\|\boldsymbol{\theta}\|_{p^*},$$

with the last inequality due to the fact that $\frac{\exp\left(-y(\boldsymbol{x}^\top\boldsymbol{\theta})\right)}{1 + \exp\left(-y(\boldsymbol{x}^\top\boldsymbol{\theta})\right)} < 1$. Taking the expectation of the expression gives:

$$F^{\mathrm{adv}}(\boldsymbol{\theta}; \delta) \leq F(\boldsymbol{\theta}; \delta) + \delta\|\boldsymbol{\theta}\|_{p^*}$$

□

**Main results**. We now present the proofs for Theorem 3.1 and 3.2 and Corollary 3.3.

*Proof of Theorem 3.1.* For brevity we will omit $\boldsymbol{\nu}$ as a subscript, such that $\epsilon = \epsilon_{\boldsymbol{\nu}}, k = k_{\boldsymbol{\nu}}, \beta = \beta_{\boldsymbol{\nu}}$.

For **(a)**, we assume $\boldsymbol{\nu}$ is in a descending order w.l.o.g., and $\hat{\boldsymbol{\nu}}$ is the corresponding vector of $\ell_2$ norms for each row. We note that

$$\|\boldsymbol{\nu}^{(k)}\|_1 = \sum_{i=1}^{k} \nu_i \geq k\nu_k \tag{13}$$

$$\geq k(1-\beta)\nu_1 \tag{14}$$

$$(1-\epsilon)\|\boldsymbol{\nu}\|_1 \geq k(1-\beta)\nu_1 \tag{15}$$

$$\frac{(1-\epsilon)}{(1-\beta)}\frac{1}{k}\|\boldsymbol{\nu}\|_1 \geq \nu_1 \tag{16}$$

$$\frac{(1-\epsilon)}{(1-\beta)}\frac{1}{k}\|\boldsymbol{\nu}\|_1 \geq \|\mathbf{W}\|_\infty \tag{17}$$

with (13) being the smallest magnitude element in $\boldsymbol{\nu}^{(k)}$, (14) due to the definition of slack variable $\beta$, and (15) due to Lemma A.1, and (17) due to the fact that $\|\mathbf{W}\|_\infty = \nu_1$, as $\boldsymbol{\nu}$ is assumed to be magnitude-ordered. We then move on to characterizing $\|\boldsymbol{\nu}\|_1$. Notice that

$$\|\boldsymbol{\nu}\|_1 = \sum_{i=1}^{h} \nu_i \leq \sum_{i=1}^{h} \sqrt{h}\hat{\nu}_i \tag{18}$$

$$\leq \sqrt{h}\|\hat{\boldsymbol{\nu}}\|_1 \tag{19}$$

$$\leq \sqrt{h}\left(\sqrt{k_r}\|\hat{\boldsymbol{\nu}}^{(k_r)}\|_2 + \sqrt{h}\|\hat{\boldsymbol{\nu}}\|_2\right) \tag{20}$$

$$\leq \left(\sqrt{hk_r} + \sqrt{h}\epsilon_r\right)\|\hat{\boldsymbol{\nu}}\|_2 \tag{21}$$

$$\leq \left(\sqrt{hk_r} + \sqrt{h}\epsilon_r\right)\|\mathbf{W}\|_F \tag{22}$$

Note that (18) is due to standard norm inequality between $\ell_1$ and $\ell_2$ rows, (20) is due to Lemma A.2, and (22) is due to $\ell_2$ norm of the vector of row $\ell_2$ rows equals the Frobenius norm. Plugging (22) back into (17) gives the desired result.

For **(b)** the proof follows similarly through steps (13)-(16) by replacing $\boldsymbol{\nu}$ with $\boldsymbol{\sigma}$. After that, we continue with

$$\frac{(1-\epsilon)}{(1-\beta)}\frac{1}{k}\|\boldsymbol{\sigma}\|_1 \geq \sigma_1 \tag{23}$$

$$\frac{(1-\epsilon)}{(1-\beta)}\frac{1}{k}\|\boldsymbol{\sigma}\|_1 \geq \|\mathbf{W}\|_2 \tag{24}$$

$$\frac{(1-\epsilon)}{(1-\beta)}\frac{\sqrt{h}}{k}\|\boldsymbol{\sigma}\|_2 \geq \|\mathbf{W}\|_2 \tag{25}$$

$$\frac{(1-\epsilon)}{(1-\beta)}\frac{\sqrt{h}}{k}\|\mathbf{W}\|_F \geq \|\mathbf{W}\|_2 \tag{26}$$

with (24) due to $\|\mathbf{W}\|_2 = \sigma_1$, (25) due to standard norm inequality between $\ell_1$ and $\ell_2$ norms, and (26) due to the fact that $\ell_2$ norm of singular values equals Frobenius norm, i.e. $\|\mathbf{W}\|_F = \|\boldsymbol{\sigma}\|_2$. $\qquad\square$

*Proof of Theorem 3.2.* Proofs for both conditions rely on an additive decomposition of the layer matrices $\mathbf{W}^l$ into dominant/leading terms vs. remainder terms, i.e. $\mathbf{W}^l = \mathbf{W}_k^l + \mathbf{W}_r^l$. In structured compressibility this takes the form of $\mathbf{W}_k^l$ and $\mathbf{W}_r^l$ including $k$ leading (largest $\ell_1$ norm) rows and $h - k$ remaining rows, respectively, with the rest of the rows set to $\mathbf{0}$ in both cases. In spectral compressibility, this takes the form of $\mathbf{W}_k^l + \mathbf{W}_r^l = \mathbf{U}_k^l\boldsymbol{\Sigma}_k^l(\mathbf{V}_k^l)^\top + \mathbf{U}_r^l\boldsymbol{\Sigma}_r^l(\mathbf{V}_r^l)^\top$, where the remaining $h - k$ vs. leading $k$ singular values are set to 0 respectively.

Let $\boldsymbol{z}^l$ denote the post-activation representations of the network after layer $l \in [\lambda]$. The Jacobian of the network output $\boldsymbol{z}^\lambda$ with respect to the input $\boldsymbol{x}$ is given by:

$$\mathbf{J}_\Phi(\boldsymbol{x}) = \mathbf{D}^\lambda(\boldsymbol{x})\mathbf{W}^\lambda\mathbf{D}^{\lambda-1}(\boldsymbol{x})\mathbf{W}^{\lambda-1}\mathbf{D}^{\lambda-2}(\boldsymbol{x})\ldots\mathbf{D}^1(\boldsymbol{x})\mathbf{W}^1, \tag{27}$$

where $\mathbf{D}^l(\boldsymbol{x})$ is the diagonal binary matrix corresponding to the ReLU activation after layer $l$, i.e., $(\mathbf{D}^l)_{ii} = \mathbb{I}[(\bar{\boldsymbol{z}}^l)_i > 0]$, with $\bar{\boldsymbol{z}}^l$ being the pre-activation representation at layer $l$ for input $\boldsymbol{x}$.

Letting $L_\Phi^p$ denote the $p$-norm Lipschitz constant of the encoder in the input domain, it can be computed as the maximum $p \to p$ operator norm of the Jacobian over the input space $\mathcal{X}$:

$$L_\Phi^p = \sup_{\boldsymbol{x} \in \mathcal{X}} \|\mathbf{J}_\Phi(\boldsymbol{x})\|_p = \sup_{\boldsymbol{x} \in \mathcal{X}} \|\mathbf{D}^\lambda(\boldsymbol{x})\mathbf{W}^\lambda \mathbf{D}^{\lambda-1}(\boldsymbol{x})\mathbf{W}^{\lambda-1} \ldots \mathbf{D}^1(\boldsymbol{x})\mathbf{W}^1\|_p. \tag{28}$$

For brevity, we use the following notation:

$$\mathbf{P}(\mathbf{D}) := \mathbf{D}^\lambda(\boldsymbol{x})\mathbf{W}^\lambda \mathbf{D}^{\lambda-1}(\boldsymbol{x})\mathbf{W}^{\lambda-1} \ldots \mathbf{D}^1(\boldsymbol{x})\mathbf{W}^1. \tag{29}$$

Note that the optimization over $\mathcal{X}$ can be upper bounded by the optimization over all binary activation matrices $\mathbf{D}^l \in \mathcal{D}$ for each layer whenever convenient. We replace the notation $\mathbf{D}^l(\boldsymbol{x})$ with $\mathbf{D}^l$ when doing so.

Note that in this proof, for increased precision and brevity we introduce the following notation for the interlayer alignment terms:

$$A_{p,l}^* := \max_{\mathbf{D} \in \mathcal{D}} A_{p,l} \tag{30}$$

where $A_{p,l}$ stands for the inner RHS term optimized over in (6).

**(a) Row/neuron compressibility** We aim to bound $L_\Phi^\infty$ as:

$$L_\Phi^\infty \le \max_{\mathbf{D}^1,\ldots,\mathbf{D}^\lambda} \|\mathbf{P}(\mathbf{D})\|_\infty. \tag{31}$$

We start by noting that we can upper bound this norm by partitioning the inside terms based on the submultiplicative property:

$$\|\mathbf{P}(\mathbf{D})\|_\infty \le \|\mathbf{D}^\lambda \mathbf{W}^\lambda \mathbf{D}^{\lambda-1} \mathbf{W}^{\lambda-1} \ldots \mathbf{D}^1 \mathbf{W}^1\|_\infty \tag{32}$$

$$\le \|\mathbf{W}^\lambda \mathbf{D}^{\lambda-1} \mathbf{W}^{\lambda-1}\|_\infty \|\mathbf{D}^{\lambda-2}\|_\infty \|\mathbf{W}^{\lambda-2}\|_\infty$$

$$\ldots \|\mathbf{W}^{l+1} \mathbf{D}^l \mathbf{W}^1\|_\infty \ldots \|\mathbf{D}^1\|_\infty \|\mathbf{W}^1\|_\infty \tag{33}$$

Note that any such parsing is valid as long as a layer does not appear in two interlayer terms at once. Given a valid parsing set $S \subseteq \{1, 2, \ldots, \lambda-1\}$, we have the interlayer alignment terms for $l \in S$, i.e. $\|\mathbf{W}^{l+1}\mathbf{D}^l\mathbf{W}^l\|_\infty$ and standalone terms for all remaining layers $\{l \mid l \notin S, l+1 \notin S\}$: $\|\mathbf{W}^l\|_\infty$. We denote all such valid parsing layer subsets with $\mathcal{S}$, where $S$ does not include any consecutive indices for any $S \in \mathcal{S}$. We will first prove the bound for any valid parsing set, and then define the optimal alignment parsing set that would lead to the tightest bound.

We first analyze a generic alignment term, using the additive decomposition into leading and remainder terms. Remember that for layer $l$ we denote the row $\ell_1$ norms with $\boldsymbol{\nu}^l = (\nu_1^l, \ldots, \nu_h^l)$, and w.l.o.g. assume that the rows are ordered in descending order according to $\nu_l$. Also note that $\|\mathbf{W}_k^l\|_\infty = \|\mathbf{W}^l\|_\infty = \nu_1^l$.

$$\|\mathbf{W}^{l+1}\mathbf{D}^l\mathbf{W}^l\|_\infty \le \|\mathbf{W}_k^{l+1}\mathbf{D}^l\mathbf{W}_k^l\|_\infty + \|\mathbf{W}_k^{l+1}\mathbf{D}^l\mathbf{W}_r^l\|_\infty$$

$$+ \|\mathbf{W}_r^{l+1}\mathbf{D}^l\mathbf{W}_k^l\|_\infty + \|\mathbf{W}_r^{l+1}\mathbf{D}^l\mathbf{W}_r^l\|_\infty \tag{34}$$

$$\le \|\mathbf{W}_k^{l+1}\mathbf{D}^l\mathbf{W}_k^l\|_\infty + \|\mathbf{W}_k^{l+1}\|_\infty\|\mathbf{W}_r^l\|_\infty$$

$$+ \|\mathbf{W}_r^{l+1}\|_\infty\|\mathbf{W}_k^l\|_\infty + \|\mathbf{W}_r^{l+1}\|_\infty\|\mathbf{W}_r^l\|_\infty \tag{35}$$

$$\le \|\mathbf{W}^{l+1}\|_\infty\|\mathbf{W}^l\|_\infty\Big(\frac{\|\mathbf{W}_k^{l+1}\mathbf{D}^l\mathbf{W}_k^l\|_\infty}{\|\mathbf{W}^{l+1}\|_\infty\|\mathbf{W}^l\|_\infty} + \frac{\nu_{k+1}^l}{\nu_1^l}$$

$$+ \frac{\nu_{k+1}^{l+1}}{\nu_1^{l+1}} + \frac{\nu_{k+1}^l \nu_{k+1}^{l+1}}{\nu_1^l \nu_1^{l+1}}\Big). \tag{36}$$

$$\le \|\mathbf{W}^{l+1}\|_\infty\|\mathbf{W}^l\|_\infty\left(\frac{\|\mathbf{W}_k^{l+1}\mathbf{D}^l\mathbf{W}_k^l\|_\infty}{\|\mathbf{W}^{l+1}\|_\infty\|\mathbf{W}^l\|_\infty} + R_\infty(\epsilon)\right). \tag{37}$$

Since the remaining, standalone layer norms also contribute $\|\mathbf{W}^l\|_\infty$, we have

$$\|\mathbf{P}(\mathbf{D})\|_\infty \le \prod_{l=1}^\lambda \|\mathbf{W}^l\|_\infty \prod_{l \in S}\left(\frac{\|\mathbf{W}_k^{l+1}\mathbf{D}^l\mathbf{W}_k^l\|_\infty}{\|\mathbf{W}^{l+1}\|_\infty\|\mathbf{W}^l\|_\infty} + R_\infty(\epsilon)\right). \tag{38}$$

Bounding the Lipschitz constant accordingly:

$$L_\Phi^\infty \le \max_{\mathbf{D}^1,\ldots,\mathbf{D}^\lambda} \prod_{l=1}^\lambda \|\mathbf{W}^l\|_\infty \prod_{l=1}^{\lambda-1}\left(\frac{\|\mathbf{W}_k^{l+1}\mathbf{D}^l\mathbf{W}_k^l\|_\infty}{\|\mathbf{W}^{l+1}\|_\infty\|\mathbf{W}^l\|_\infty} + R_\infty(\epsilon)\right) \tag{39}$$

$$= \prod_{l=1}^\lambda \|\mathbf{W}^l\|_\infty \prod_{l \in S}\left(\max_{\mathbf{D} \in \mathcal{D}}\frac{\|\mathbf{W}_k^{l+1}\mathbf{D}\mathbf{W}_k^l\|_\infty}{\|\mathbf{W}^{l+1}\|_\infty\|\mathbf{W}^l\|_\infty} + R_\infty(\epsilon)\right) \tag{40}$$

$$= \prod_{l=1}^\lambda \|\mathbf{W}^l\|_\infty \prod_{l \in S} A_\infty^*(\mathbf{W}^{l+1}, \mathbf{W}^l) + R_\infty(\epsilon). \tag{41}$$

Contributing an alignment term of 1 for $\{l \mid l \notin S, l+1 \notin S\}$ gives the desired result if $S = S_{opt}$, which we define below.

Given multiple valid parsing sets are possible whenever $\lambda > 2$, we lastly define the *optimal alignment parsing set*, $S_{opt}$.

**Definition A.4** (Optimal Alignment Parsing Set). *The Optimal Alignment Parsing Set $S_{opt}$ is a set in $\mathcal{S}$ that achieves the minimum product of the corresponding maximum alignment factors:*

$$S_{opt} \in \arg\min_{S \in \mathcal{S}} \prod_{l \in S} A^*_{\infty,l}. \tag{42}$$

*Note that $S_{opt}$ might not be unique, but $\min_{S \in \mathcal{S}} \prod_{l \in S} A^*_{\infty,l}$ is.*

**Complexity of finding $S_{opt}$:** Finding $S_{opt} \in \arg\min_{S \in \mathcal{S}} \prod_{l \in S} A^*_{\infty,l}$ is equivalent to finding the independent set $S$ in the path graph $G = (V, E)$ with $V = \{1, \ldots, \lambda - 1\}$ that maximizes $\sum_{l \in S} w_l$, where weights $w_l = -\log A^*_{\infty,l}$ (assuming $A^*_{\infty,l} > 0$; we handle $A^*_{\infty,l} = 0$ as a special case yielding $\prod_{l \in S_{opt}} A^*_{\infty,l} = 0$). This is the Maximum Weight Independent Set, which can be solved in linear time in chordal graphs, of which path graphs are a subfamily (Frank, 1976).

**(b) Spectral compressibility:** We can upper bound $L^2_\Phi$ by considering all possible activation patterns (all possible binary diagonal matrices $\mathbf{D}^l$):

$$L^2_\Phi \leq \max_{\mathbf{D}^1,\ldots,\mathbf{D}^\lambda} \|\mathbf{P}(\mathbf{D})\|_2 \tag{43}$$

We modify the SVD decomposition for layers as

$$\mathbf{W}^l = \mathbf{U}^l \sqrt{\mathbf{\Sigma}^l} \sqrt{\mathbf{\Sigma}^l} (\mathbf{V}^l)^\top \tag{44}$$

$$= \underbrace{\left(\mathbf{U}^l_k \sqrt{\mathbf{\Sigma}^l_k} + \mathbf{U}^l_r \sqrt{\mathbf{\Sigma}^l_r}\right)}_{\mathbf{A}^l} \underbrace{\left(\sqrt{\mathbf{\Sigma}^l_k}(\mathbf{V}^l_k)^\top + \sqrt{\mathbf{\Sigma}^l_r}(\mathbf{V}^l_r)^\top\right)}_{\mathbf{B}^l}. \tag{45}$$

Note that we assume untruncated singular vector matrices for $\mathbf{W}^l_k$ and $\mathbf{W}^l_r$ for the equation above to be valid. We then decompose the spectral norm using the submultiplicative property:

$$\|\mathbf{P}(\mathbf{D})\|_2 = \|\mathbf{D}^\lambda \mathbf{W}^\lambda \mathbf{D}^{\lambda-1} \mathbf{W}^{\lambda-1} \mathbf{D}^{\lambda-2} \ldots \mathbf{D}^1 \mathbf{W}^1\|_2 \tag{46}$$

$$\leq \|\mathbf{A}^\lambda\|_2 \|\mathbf{B}^\lambda \mathbf{D}^{\lambda-1} \mathbf{A}^{\lambda-1}\|_2 \|\mathbf{B}^{\lambda-1} \mathbf{D}^{\lambda-2} \mathbf{A}^{\lambda-2}\|_2$$
$$\ldots \|\mathbf{B}^{l+1} \mathbf{D}^l \mathbf{A}^l\|_2 \ldots \|\mathbf{B}^2 \mathbf{D}^1 \mathbf{A}^1\|_2 \|\mathbf{B}^1\|_2 \tag{47}$$

We then analyze the central term $\|\mathbf{B}^{l+1} \mathbf{D}^l \mathbf{A}^l\|_2$, and decompose it using the submultiplicative and subadditivity properties. Remember that for layer $l$ we denote the singular values with $\boldsymbol{\sigma}^l = (\sigma^l_1, \ldots, \sigma^l_h)$. Also note that $\|\mathbf{W}^l_k\|_2 = \|\mathbf{W}^l\|_2 = \sigma^l_1$.

$$\|\mathbf{B}^{l+1} \mathbf{D}^l \mathbf{A}^l\|_2$$

$$\leq \|\sqrt{\mathbf{\Sigma}^{l+1}_k}(\mathbf{V}^{l+1}_k)^\top \mathbf{D}^l \mathbf{U}^l_k \sqrt{\mathbf{\Sigma}^l_k}\|_2 + \|\sqrt{\mathbf{\Sigma}^{l+1}_k}(\mathbf{V}^{l+1}_k)^\top \mathbf{D}^l \mathbf{U}^l_r \sqrt{\mathbf{\Sigma}^l_r}\|_2$$

$$+ \|\sqrt{\mathbf{\Sigma}^{l+1}_r}(\mathbf{V}^{l+1}_r)^\top \mathbf{D}^l \mathbf{U}^l_k \sqrt{\mathbf{\Sigma}^l_k}\|_2 + \|\sqrt{\mathbf{\Sigma}^{l+1}_r}(\mathbf{V}^{l+1}_r)^\top \mathbf{D}^l \mathbf{U}^l_r \sqrt{\mathbf{\Sigma}^l_r}\|_2 \tag{48}$$

$$\leq \|\sqrt{\mathbf{\Sigma}^{l+1}_k}(\mathbf{V}^{l+1}_k)^\top \mathbf{D}^l \mathbf{U}^l_k \sqrt{\mathbf{\Sigma}^l_k}\|_2 + \sqrt{\sigma^{l+1}_1}\|(\mathbf{V}^{l+1}_k)^\top \mathbf{D}^l \mathbf{U}^l_r\|_2 \sqrt{\sigma^l_{k+1}}$$

$$+ \sqrt{\sigma^{l+1}_{k+1}}\|(\mathbf{V}^{l+1}_r)^\top \mathbf{D}^l \mathbf{U}^l_r\|_2 \sqrt{\sigma^l_1} + \sqrt{\sigma^{l+1}_{k+1}}\|(\mathbf{V}^{l+1}_r)^\top \mathbf{D}^l \mathbf{U}^l_r\|_2 \sqrt{\sigma^l_{k+1}} \tag{49}$$

$$\leq \sqrt{\sigma^{l+1}_1}\sqrt{\sigma^l_1}\left(\frac{\|\sqrt{\mathbf{\Sigma}^{l+1}_k}(\mathbf{V}^{l+1}_k)^\top \mathbf{D}^l \mathbf{U}^l_k \sqrt{\mathbf{\Sigma}^l_k}\|_2}{\sqrt{\sigma^l_1 \sigma^{l+1}_1}} + \sqrt{\frac{\sigma^l_{k+1}}{\sigma^l_1}} + \sqrt{\frac{\sigma^{l+1}_{k+1}}{\sigma^l_1}} + \sqrt{\frac{\sigma^l_{k+1}\sigma^{l+1}_{k+1}}{\sigma^l_1 \sigma^{l+1}_1}}\right) \tag{50}$$

$$\leq \sqrt{\sigma^{l+1}_1}\sqrt{\sigma^l_1}\left(\frac{\|\sqrt{\mathbf{\Sigma}^{l+1}_k}(\mathbf{V}^{l+1}_k)^\top \mathbf{D}^l \mathbf{U}^l_k \sqrt{\mathbf{\Sigma}^l_k}\|_2}{\sqrt{\sigma^l_1 \sigma^{l+1}_1}} + R_2(\epsilon)\right), \tag{51}$$

where we set all cross-alignment terms other than dominant-dominant interaction to 1. This is made possible by the fact that they are the multiplication of orthogonal matrices and a ReLU matrix, all of which have spectral

norms upper bounded by 1. Note that for all layers $l \in 1, \dots, \lambda$, $\sqrt{\sigma_1^l}$ will appear twice in the multiplication, including the first and last layers due to the leading and final terms in (47), leading to the expression:

$$\|\mathbf{P}(\mathbf{D})\|_2 \leq \prod_{l=1}^{\lambda} \|\mathbf{W}^l\|_2 \prod_{l=1}^{\lambda-1} \left( \frac{\|\sqrt{\boldsymbol{\Sigma}_k^{l+1}}(\mathbf{V}_k^{l+1})^\top \mathbf{D}^l \mathbf{U}_k^l \sqrt{\boldsymbol{\Sigma}_k^l}\|_2}{\sqrt{\sigma_1^l \sigma_1^{l+1}}} + R_2(\epsilon) \right) \tag{52}$$

Bounding the Lipschitz constant:

$$L_\Phi^2 \leq \max_{\mathbf{D}^1, \dots, \mathbf{D}^\lambda} \|\mathbf{P}(\mathbf{D})\|_2 \tag{53}$$

$$\leq \max_{\mathbf{D}^1, \dots, \mathbf{D}^\lambda} \prod_{l=1}^{\lambda} \|\mathbf{W}^l\|_2 \prod_{l=1}^{\lambda-1} \left( \frac{\|\sqrt{\boldsymbol{\Sigma}_k^{l+1}}(\mathbf{V}_k^{l+1})^\top \mathbf{D}^l \mathbf{U}_k^l \sqrt{\boldsymbol{\Sigma}_k^l}\|_2}{\sqrt{\sigma_1^l \sigma_1^{l+1}}} + R_2(\epsilon) \right) \tag{54}$$

$$\leq \prod_{l=1}^{\lambda} \|\mathbf{W}^l\|_2 \prod_{l=1}^{\lambda-1} \left( \max_{\mathbf{D} \in \mathcal{D}} \frac{\|\sqrt{\boldsymbol{\Sigma}_k^{l+1}}(\mathbf{V}_k^{l+1})^\top \mathbf{D}^l \mathbf{U}_k^l \sqrt{\boldsymbol{\Sigma}_k^l}\|_2}{\sqrt{\sigma_1^l \sigma_1^{l+1}}} + R_2(\epsilon) \right) \tag{55}$$

$$\leq \prod_{l=1}^{\lambda} \|\mathbf{W}^l\|_2 \prod_{l=1}^{\lambda-1} A_2^*(\mathbf{W}_k^{l+1}, \mathbf{W}_k^l), \tag{56}$$

yielding the desired result.

$\square$

*Proof of Corollary 3.3.* Let $\boldsymbol{a}$ denote the adversarial perturbation on the input $\boldsymbol{x}$, where $\|\boldsymbol{a}\|_p \leq \delta$. We define the *effective perturbation budget* in $\ell_p$ norm for the feature encoder $\Phi$ as $\delta_\Phi^p := \max \|\Phi(x) - \Phi(x+a)\|_p$. Note that by definition of the Lipschitz constant and by Theorem 3.2, we have

$$\delta_\Phi^p = \max \|\Phi(\boldsymbol{x}) - \Phi(\boldsymbol{x}+\boldsymbol{a})\|_p \leq \|\boldsymbol{x} - (\boldsymbol{x}+\boldsymbol{a})\|_p L_\Phi^p \leq \|\boldsymbol{a}\|_p \tilde{L}_\Phi^p = \delta \tilde{L}_\Phi^p. \tag{57}$$

Plugging the result back into (12) yields the desired result. Note that for clarity, the corollary uses $\mathbf{C}$ in the main text, instead of $\boldsymbol{\theta}$ in (12). $\square$

**Lemma A.5.** *Under the conditions described in Theorem 3.2, $R_p \to 0$ as $\epsilon \to 0$ for $p \in \{2, \infty\}$.*

*Proof.* $p = \infty$: Due to the definition of compressibility, for all $l \in [\lambda]$,

$$\|\boldsymbol{\nu}^l - \boldsymbol{\nu}_k^l\|_1 \leq \epsilon \|\boldsymbol{\nu}^l\|_1 \tag{58}$$

$$\nu_{k+1}^l \leq \epsilon h \|\mathbf{W}^l\|_F, \tag{59}$$

by applying standard norm inequalities across rows and columns. The result follows from noting that the final inequality applies to both $\nu_{k+1}^l$ and $\nu_{k+1}^{l+1}$.

$p = 2$: Similarly, due to the definition of compressibility, for all $l \in [\lambda]$,

$$\|\boldsymbol{\sigma}^l - \boldsymbol{\sigma}_k^l\|_1 \leq \epsilon \|\boldsymbol{\sigma}^l\|_1 \tag{60}$$

$$\sigma_{k+1}^l \leq \epsilon \sqrt{h} \|\mathbf{W}^l\|_F, \tag{61}$$

since $\|\boldsymbol{\sigma}^l\|_2 = \|W^l\|_F$. The result follows from noting that the final inequality applies to both $\sigma_{k+1}^l$ and $\sigma_{k+1}^{l+1}$. $\square$

**Lemma A.6.** *Under the conditions described in Theorem 3.2, $A_p^*(\mathbf{W}^{l+1}, \mathbf{W}^l)$ can be replaced with $\min(1, A_p^*(\mathbf{W}^{l+1}, \mathbf{W}^l))$ for $p \in \{2, \infty\}$.*

*Proof.* For $p = \infty$,

$$\max_{\mathbf{D} \in \mathcal{D}} \frac{\|\mathbf{W}^{l+1} \mathbf{D} \mathbf{W}^l\|_\infty}{\|\mathbf{W}^{l+1}\|_\infty \|\mathbf{W}^l\|_\infty} \leq \frac{\|\mathbf{W}^{l+1}\|_\infty \max_{\mathbf{D} \in \mathcal{D}} \|\mathbf{D}\|_\infty \|\mathbf{W}^l\|_\infty}{\|\mathbf{W}^{l+1}\|_\infty \|\mathbf{W}^l\|_\infty} \tag{62}$$

$$\leq \frac{\|\mathbf{W}^{l+1}\|_\infty \|\mathbf{W}^l\|_\infty}{\|\mathbf{W}^{l+1}\|_\infty \|\mathbf{W}^l\|_\infty} = 1. \tag{63}$$

The proof follows identically for $p = 2$. $\square$

# B ADDITIONAL TECHNICAL RESULTS AND ANALYSES

## B.1 $(q, k, \epsilon)$-COMPRESSIBILITY VS. OTHER NOTIONS OF APPROXIMATE SPARSITY

**Further discussion of $(q, k, \epsilon)$-compressibility**.

Our concept of compressibility can be thought of as the generalization of *sparsity*, with the obvious advantage of being applicable to domains where true sparsity is rare, such as neural network parameter values. Note that our intuitive definition of compressibility is based on foundational results in compressed sensing and is well exploited in the established machine learning literature (Amini et al., 2011; Gribonval et al., 2012; Barsbey et al., 2021; Diao et al., 2023; Wan et al., 2024). More specifically, when $k \ll d$ and $\epsilon \ll 1$, Definition 2.1 is equivalent to Gribonval et al. (2012)'s definition of *compressible vector*. Inspired by desiderata from an ideal metric of sparsity in the economics literature, Diao et al. (2023) recently introduced another scale-invariant notion of approximate sparsity:

**Definition B.1** (PQ Index Diao et al. (2023)). *For any $0 < p < q$, the PQ Index of a non-zero vector $\mathbf{w} \in \mathbb{R}^d$ is*

$$I_{p,q}(\mathbf{w}) = 1 - d^{\frac{1}{q} - \frac{1}{p}} \frac{\|\mathbf{w}\|_p}{\|\mathbf{w}\|_q}. \tag{64}$$

Interestingly, it is possible to directly relate this notion of sparsity to $(q, k, \epsilon)$-compressibility, as shown in the following proposition.

**Proposition B.2.** *Given $0 < p < q$, for a vector $\boldsymbol{\theta}$, its $(q, k, \epsilon)$ compressibility implies the following lower bound for its PQ Index:*

$$1 - \epsilon - \kappa^\phi \le I_{p,q}(\boldsymbol{\theta}), \tag{65}$$

*where $\kappa = k/d$ and $\phi = \frac{1}{p} - \frac{1}{q}$. Note that the constraints on $p, q$ imply $\phi > 0$.*

*Proof.* Let $\gamma = \frac{1}{p} - \frac{1}{q}$. Note that from (11) we know that $\|\boldsymbol{\theta}\|_p \le (k^\gamma + d^\gamma \epsilon) \|\boldsymbol{\theta}\|_q$. This implies

$$\frac{\|\boldsymbol{\theta}\|_p}{\|\boldsymbol{\theta}\|_q} \le k^\gamma + d^\gamma \epsilon. \tag{66}$$

Note that PQ Index from (64) can be written as $(1 - I_{p,q}(\boldsymbol{\theta}))d^\gamma = \frac{\|\boldsymbol{\theta}\|_p}{\|\boldsymbol{\theta}\|_q}$. Plugging this into the LHS of (66) and simple algebraic manipulation gives the desired result. $\square$

*Remark* B.3. Assume that $\boldsymbol{\theta}$ and $\boldsymbol{\theta}'$ are $(q, k, \epsilon)$ and $(q, k', \epsilon')$ compressible respectively. If $k = k'$ and $\epsilon < \epsilon'$; or $k < k'$ and $\epsilon = \epsilon'$ implies a larger lower bound on PQI. That is, a larger $(q, k, \epsilon)$ compressibility suggests a larger PQI.

**Dominance vs. spread**. While $(q, k, \epsilon)$-compressibility quantifies how well a vector can be approximated using its top-$k$ entries (*e.g.* top-$k$ filters or singular values), it does not fully capture the internal structure among those dominant terms. Consider the vectors $\boldsymbol{x}_1 = (10, 2, 1, 1)$ and $\boldsymbol{x}_2 = (6, 6, 1, 1)$: both yield the same 2-term relative approximation error under $q = 1$, yet their dominant components differ markedly in structure. To formalize this distinction, we introduce the **spread variable** as a complementary descriptor. Given a vector $\boldsymbol{\theta}$ with elements sorted by magnitude, we define its *spread* $\beta \in [0, 1]$ via the relation $|\theta_k| = (1 - \beta)|\theta_1|$. Intuitively, $\beta$ quantifies the relative decay from the largest to the $k$-th largest entry, capturing an additional degree of freedom in the geometry of compressibility, better describing and distinguishing compressible distributions beyond what is possible with approximation error alone.

Lastly, to provide a numerical comparison, consider $\boldsymbol{x}_1 = (6.00, 1.50, 0.75, 0.75)$ and $\boldsymbol{x}_2 = (4.00, 4.00, 0.057, 0.057)$. The qualitative difference between the two vectors is obvious, and is easy to observe under our compressibility definition: with $q = 2, k = 2$, we have $\epsilon = 0.169, \beta = 0.75$ vs. $\epsilon = 0.014, \beta = 0.00$, respectively. Note that this difference is captured neither by the classical notion of sparsity (neither vector includes any 0 elements), nor the more modern PQ Index, as both vectors have a PQI$(2, 1)$ of 0.697.

## B.2 LOWER BOUNDS ON OPERATOR NORMS

The following theorem characterizes the compressibility-based lower bounds of operator norms, complementing the upper bounds presented in the main paper.

**Theorem B.4.** *The following statements lower bound operator norms using compressibility and Frobenius norm.*

*(a) Neuron compressibility (i.e. row-sparsity):* Let $\mathbf{w}_i, i \in [h]$ denote the rows of the matrix $\mathbf{W}$, and let $\boldsymbol{\nu} := (\|\mathbf{w}_1\|_1, \ldots, \|\mathbf{w}_h\|_1)$ denote $\ell_1$ norms of its rows. Assuming $\boldsymbol{\nu}$ is $(1, k_{\boldsymbol{\nu}}, \epsilon_{\boldsymbol{\nu}})$ and each row $\mathbf{w}_i$ is $(2, k_r, \epsilon_r)$ compressible implies:

$$\left( \frac{\sqrt{k_r}}{\sqrt{k_r(1 - \epsilon_r^2)} + \sqrt{\epsilon_r}} \right) \frac{(1 - \epsilon_{\boldsymbol{\nu}})}{k_{\boldsymbol{\nu}}} \|\mathbf{W}\|_F \le \|\mathbf{W}\|_\infty. \tag{67}$$

*(b) Spectral compressibility (i.e. low-rankness):* Let $\boldsymbol{\sigma} := (\sigma_1, \sigma_2, \ldots)$ denote the singular values of matrix $\mathbf{W}$. Assuming $\boldsymbol{\sigma}$ is $(1, k_{\boldsymbol{\sigma}}, \epsilon_{\boldsymbol{\sigma}})$ compressible implies:

$$\sqrt{\frac{(1 - h\epsilon_{\boldsymbol{\sigma}}^2)}{k_{\boldsymbol{\sigma}}}} \|\mathbf{W}\|_F \le \|\mathbf{W}\|_2. \tag{68}$$

*Proof.* For **(a)** note that $\|\mathbf{W}\|_\infty = \|\boldsymbol{\nu}\|_\infty$. Note that the minimum value this value can take is $\|\boldsymbol{\nu}_k\|_1 / k_{\boldsymbol{\nu}}$. By the definition of strict compressibility, we know that $\|\boldsymbol{\nu}_k\|_1 = (1 - \epsilon)\|\boldsymbol{\nu}\|_1$. This gives us the inequality:

$$\frac{(1 - \epsilon_{\boldsymbol{\nu}})}{k_{\boldsymbol{\nu}}} \|\boldsymbol{\nu}\|_1 \le \|\mathbf{W}\|_\infty. \tag{69}$$

We then turn to the components of $\boldsymbol{\nu}$, and examine the relationship between $\|\mathbf{w}\|_2$ and $\|\mathbf{w}\|_1$ for any row $\mathbf{w}$. We will use $\mathbf{w}_k$, $\mathbf{w}_r$ to refer to the dominant and remainder terms of $\mathbf{w}$ respectively. We invoke Minkowski's inequality:

$$\|\mathbf{w}\|_2 \le \|\mathbf{w}_k\|_2 + \|\mathbf{w}_r\|_2. \tag{70}$$

We bound the leftmost term by $\|\mathbf{w}_k\|_2 \le \sqrt{1 - \epsilon_r^2}\|\mathbf{w}\|_2 \le \sqrt{1 - \epsilon_r^2}\|\mathbf{w}\|_1$ due to Lemma A.1. For the term $\|\mathbf{w}_r\|_2$, we observe that due to interpolation inequality:

$$\|\mathbf{w}_r\|_2 \le \|\mathbf{w}_r\|_1^{\frac{1}{2}} \|\mathbf{w}_r\|_\infty^{\frac{1}{2}}. \tag{71}$$

Examining $\|\mathbf{w}_r\|_\infty$, we note that the maximum magnitude $\mathbf{w}_r$ can contain is less than or equal to the maximum value the lowest magnitude element of $\mathbf{w}_k$ can take. This is the case when all elements of $\mathbf{w}_k$ are equal, therefore $\|\mathbf{w}_r\|_\infty \le \|\mathbf{w}_k\|_1 / k$. Using this, the fact that $\|\mathbf{w}_k\|_1 \le \|\mathbf{w}\|_1$, and that $\|\mathbf{w}_r\|_1 \le \epsilon\|\mathbf{w}\|_1$ by compressibility definition, we can write:

$$\|\mathbf{w}_r\|_2 \le \|\mathbf{w}_r\|_1^{\frac{1}{2}} \|\mathbf{w}_r\|_\infty^{\frac{1}{2}} \le \epsilon^{\frac{1}{2}} \|\mathbf{w}\|_1^{\frac{1}{2}} \left( \frac{\|\mathbf{w}\|_1}{k} \right)^{\frac{1}{2}} \le \frac{\sqrt{\epsilon}}{\sqrt{k}} \|\mathbf{w}\|_1,$$

Plugging this back into the additive decomposition of $\|\mathbf{w}\|_2$ above, we have:

$$\frac{\sqrt{k}}{\sqrt{k(1 - \epsilon^2)} + \sqrt{\epsilon}} \|\mathbf{w}\|_2 \le \|\mathbf{w}\|_1. \tag{72}$$

Let $\hat{\boldsymbol{\nu}}$ denote the $\ell_2$ norms of $\mathbf{W}$'s rows. Then, plugging this back to the main inequality:

$$\|\mathbf{W}\|_\infty \ge \frac{(1 - \epsilon_{\boldsymbol{\nu}})}{k_{\boldsymbol{\nu}}} \|\boldsymbol{\nu}\|_1. \tag{73}$$

$$\ge \frac{\sqrt{k}}{\sqrt{k(1 - \epsilon^2)} + \sqrt{\epsilon}} \frac{(1 - \epsilon_{\boldsymbol{\nu}})}{k_{\boldsymbol{\nu}}} \|\hat{\boldsymbol{\nu}}\|_1 \tag{74}$$

$$\ge \frac{\sqrt{k}}{\sqrt{k(1 - \epsilon^2)} + \sqrt{\epsilon}} \frac{(1 - \epsilon_{\boldsymbol{\nu}})}{k_{\boldsymbol{\nu}}} \|\hat{\boldsymbol{\nu}}\|_2 \tag{75}$$

$$\ge \frac{\sqrt{k}}{\sqrt{k(1 - \epsilon^2)} + \sqrt{\epsilon}} \frac{(1 - \epsilon_{\boldsymbol{\nu}})}{k_{\boldsymbol{\nu}}} \|\mathbf{W}\|_F \tag{76}$$

which gives use the desired inequality.

For **(b)**, we will use $\boldsymbol{\sigma}_k$, $\boldsymbol{\sigma}_r$ to refer to the dominant and remainder terms of $\boldsymbol{\sigma}$ respectively. Note that $\|\mathbf{W}\|_F^2 = \|\boldsymbol{\sigma}\|_2^2 = \|\boldsymbol{\sigma}_k\|_2^2 + \|\boldsymbol{\sigma}_r\|_2^2$. We bound the norm of the dominant singular values by $\|\boldsymbol{\sigma}_k\|_2^2 \le k\sigma_1^2 = k\|\mathbf{W}\|_2^2$. We bound the remainder singular values by noting that

$$\|\boldsymbol{\sigma}_r\|_2^2 \le (\|\boldsymbol{\sigma}_r\|_1)^2 \le (\epsilon_{\boldsymbol{\sigma}}\|\boldsymbol{\sigma}\|_1)^2 \le \epsilon_{\boldsymbol{\sigma}}^2(\sqrt{h}\|\boldsymbol{\sigma}\|_2)^2 = h\epsilon_{\boldsymbol{\sigma}}^2\|\mathbf{W}\|_F^2. \tag{77}$$

This gives us the inequality:

$$\|\mathbf{W}\|_F^2 \le k\|\mathbf{W}\|_2^2 + h\epsilon_{\boldsymbol{\sigma}}^2\|\mathbf{W}\|_F^2. \tag{78}$$

Rearranging the terms gives the desired lower bound. $\qquad\square$

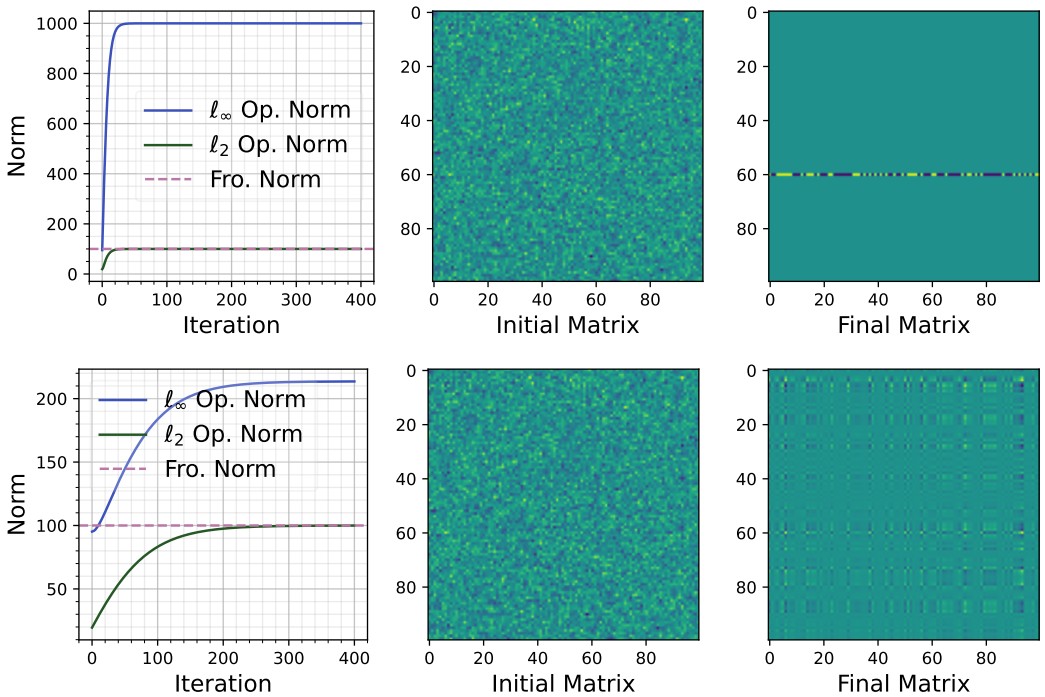

Figure 9: Optimizing for $\ell_\infty$ (top) and $\ell_2$ (bottom) operator norms.

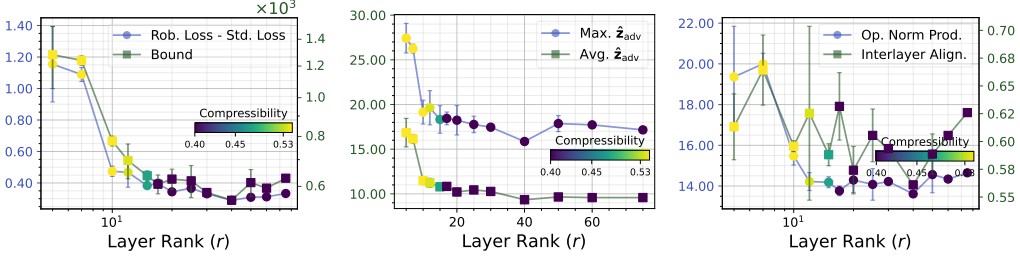

Figure 10: Empirically investigating the implications of Theorem 3.2.

### B.3 RELATIONSHIPS BETWEEN OPERATOR NORMS

Although Theorem 3.1 directly relates $\ell_\infty$ and $\ell_2$ operator norms to neuron and spectral compressibility, both the known norm inequality relationships and our results on cross-norm adversarial attacks imply that these two quantities are likely to be strongly correlated under this context. We conduct simple experiment to test this hypothesis: We optimize for either $\ell_\infty$ or $\ell_2$ operator norm of a random i.i.d. Gaussian matrix $\mathbf{A}$ where $A_{i,j} \overset{\text{i.i.d.}}{\sim} \mathcal{N}(0,1)$. We then conduct a gradient ascent-based optimization of the matrix's either $\ell_\infty$ or $\ell_2$ operator norms, while normalizing the Frobenius norm to its initialization value. In Figure 9, as an average of 10 random seeds, we show how $\ell_\infty$ and $\ell_2$ evolve while either $\ell_\infty$ (top) and $\ell_2$ (bottom) are optimized. We note that in both case both norms are strongly associated in increasing simultaneously. Note that given the inequality $\|\mathbf{A}\|_2 \leq \|\mathbf{A}\|_F$, by the end of optimization the spectral norm reaches its limit in Frobenius norm. While the left column shows the norms across iterations, center and right columns portray the qualitative differences produced by optimizing for either columns. As expected, optimizing for $\ell_\infty$ collects all energy in a single row, while optimizing for $\ell_2$ produces a 1-rank matrix.

### B.4 EMPIRICAL ANALYSES OF THE ROBUSTNESS BOUND AND RELATED QUANTITIES

In this section, we directly investigate how well our bound correlates with the adversarial robustness gap, as predicted in Corollary 3.3. In order to fully conform to the setting of Corollary 3.3, we convert the previously introduced MNIST dataset to a binary classification task by converting its labels to 0-1, by assigning 0-4 to

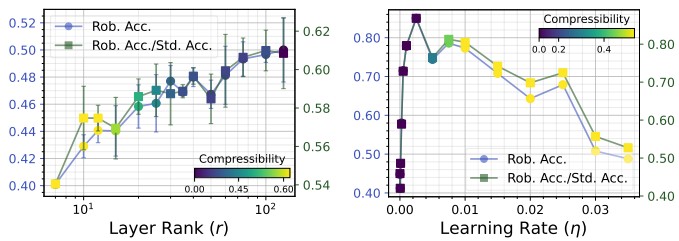

Figure 11: Adversarial fine-tuning (left) and training (center). Robust accuracy under increasing learning rate (right).

class 0 and 5-9 to class 1. We create a fully connected network (FCN) with two hidden layers of width 300, with ReLU activations after each layer. We then create networks with various spectral compressibility through varying the rank of the hidden layers, imposed through low-rank factorization. While computing the bound, we determine $k$ (num. dominant terms), and compute $\epsilon$ and $\beta$ as statistics. Note that if $\beta = 1$, this would make the bound undefined - however, instead of being a numerical problem, this implies that $k$ should be selected lower, as dominant terms including 0 is an undesired corner case. Figure 10 demonstrates the results of our experiment. First, Figure 10 (left) shows that our bound is closely correlated with adversarial robustness gap. This shows that although our bound is an order of magnitude above the empirical loss difference, it is still a faithful indicator of adversarial robustness.

We then investigate whether local input sensitivity of the network tracks its global properties. As in the main paper, letting $z = \Phi(x)$ and $z_{\mathrm{adv}} = \Phi(x + a^*)$ denote the learned representations of clean and perturbed input images, we compute $\|z - z_{\mathrm{adv}}\|_2 / \|a^*\|_2$ for 1000 test samples. We take this metric as a secant approximation of the local Lipschitz constant around input $x$. We then use the maximum and the mean of this statistic over the samples as *empirical lower bounds* to the global and expected local Lipschitz constants respectively. Figure 10 (center) shows that these two values are closely correlated with each other and with global bounds: An increase in the maximum sensitivity to perturbation is reflected in a similar increase in the average sensitivity. Lastly, Figure 10 (right) investigates the effect of spectral compressibility on interlayer alignment, in parallel to product of spectral norms of the layers (to quantify the intra- vs. interlayer dynamics in our bound). Results show that while norms increase as expected, interlayer alignment does not necessarily portray a consistent pattern. We consider how and why interlayer alignment changes in response to various compressibility inducing sparsity and training dynamics to be a crucial future research direction.

## B.5 APPROXIMATING THE INTERLAYER ALIGNMENT TERMS

Note that the interlayer alignment terms used in Theorem 3.2 lead to a combinatorial optimization problem due to the discreteness of ReLU gradients, i.e. $\{0, 1\}$. A closely related precedent from the literature is SeqLip by Scaman & Virmaux (2018), with the differences relating to the normalization of the terms, and the $k$-term adaptation. However, since these differences do not lead to any changes with respect to the optimization of these terms (*i.e.* their maxima), the authors' approximation methodology is an attractive choice for determining $A_p^*$. Scaman & Virmaux (2018) report that their gradient-ascent based greedy search algorithm is in $\sim 1\%$ of the analytical solution for cases where the latter is computationally feasible. We adopt their solution to our case for both interlayer alignment terms.

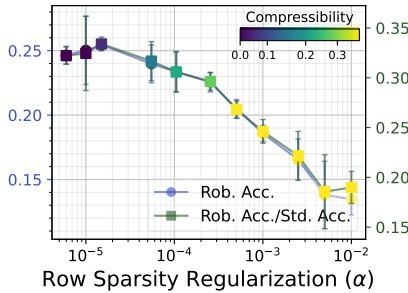

Figure 12: Effects of standard group lasso on compressibility and adversarial robustness.

## C  DETAILS OF THE EXPERIMENTAL SETTINGS AND RESOURCES

### C.1  DATASETS

Our experiments are conducted using the most commonly utilized datasets and architectures in research on adversarial robustness under pruning (Piras et al., 2025). Our datasets include MNIST (Deng, 2012), CIFAR-10, CIFAR-100 (Krizhevsky & Hinton, 2009), SVHN (Netzer et al., 2011), Flickr30k (Young et al., 2014), and ImageNet-1k (Deng et al., 2009). As detailed in Appendix B, we convert MNIST into a binary classification task for empirically investigating how our bound correlates with adversarial robustness gap. In all datasets, we use the canonical train-test splits. Whenever validation set-based model selection or early stopping is used, we utilize 10% of the training set for this task, and conduct early stopping with a patience of 10 epochs based on validation loss.

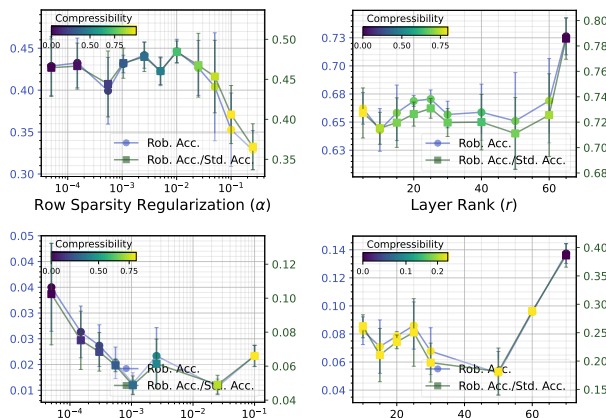

Figure 13: Results with SVHN & Wide ResNet 101-2 (top), CIFAR-100 & VGG16 (bottom).

### C.2  MODELS

Architectures we utilize include fully connected networks (FCN), ResNet18 (He et al., 2016), VGG16 (Simonyan & Zisserman, 2014), WideResNet-101-2 (Zagoruyko & Komodakis, 2016), vision transformer (ViT) - both as a standalone classifier (Dosovitskiy et al., 2021) and as part of a CLIP encoder (Radford et al., 2021), and Swin Transformer (Liu et al., 2021). Whenever needed, we apply modifications to the standard architectures in question. For our visualization experiments at the beginning of Section 4, we utilize a 1-hidden layer FCN with ReLU activation, with no bias nodes, and a width of 400. For our main results with CIFAR-10, we utilize a 2000-width FCN with 4 hidden layers, with the remaining architectural choices remain identical. Regarding the VGG16 architecture, due to our datasets being size $32 \times 32$, we remove the redundant 4096-width linear layers (along with their interleaving dropout and ReLU layers). Lastly, when conducting the low-rank factorization experiments, we modify linear layers with a factorized layer, and do the equivalent for 2D convolutional layers (Zhong et al., 2023).

For transformer models, we utilize a Base ViT architecture with $8 \times 8$ patch size. When fine-tuning a pre-trained version, we utilize a version pretrained on ImageNet-21K and fine-tuned on ImageNet-1K, hosted by the HuggingFace platform (Wolf et al., 2020). For the Swin Transformer we use a tiny version of the architecture, and utilize an ImageNet-1K pretrained version hosted by torchvision (TorchVision maintainers and contributors, 2016). For CLIP experiments, we utilize a pre-trained CLIP model, CLIP ViT-B/32, trained on LAION 2B dataset, hosted by Open CLIP (Ilharco et al., 2021). To conduct the zero-shot classification with the fine-tuned CLIP, we fine-tune the model with the Flickr30k dataset using a weight decay of 0.01 and a learning rate of $1e-5$ for 30 epochs. For the classification that follows, we present results with top-5 (standard and adversarial) accuracy, and we utilize the following prompts to embed the text descriptions, which serve as the class vectors:

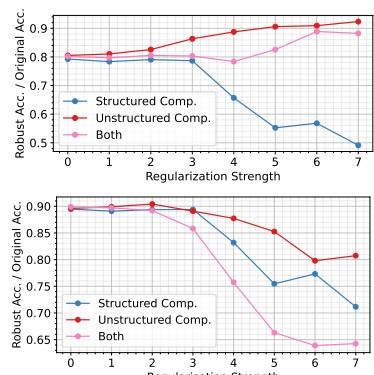

Figure 14: Unstructured alongside structured compressibility, for row/neuron (top) and spectral compressibility (bottom).

- a photo of a . . .

- a blurry photo of a . . .

- a photo of the . . .

- a close-up photo of a . . .

- a black and white photo of a . . .

- a cropped photo of a . . .

- a bright photo of a . . .

## C.3 STANDARD AND ADVERSARIAL TRAINING

**Standard training**. We normally use softmax cross-entropy loss, the AdamW optimizer with a weight decay of 0.01, a learning rate of 0.001, and use validation set based model selection for early stopping. For adversarial training tasks, we also include a cosine learning rate annealing schedule (epochs = 60, min. learning rate = 0), basic data augmentation in the form of random cropping and horizontal flips, and an adversarial validation set, again constituting 10% of the training set.

**Evaluating and training for adversarial robustness**. For evaluating adversarial robustness, we primarily employ the AutoPGD attack (Croce & Hein, 2020), using the implementation from Nicolae et al. (2018). During adversarial training, we generate adversarial examples at each iteration using the PGD attack (Madry et al., 2018). Unless stated otherwise, adversarial examples make up 50% of each training minibatch. For models trained end-to-end with adversarial robustness, we set $\epsilon = 8/255$ for $\ell_\infty$ attacks and $\epsilon = 0.5$ for $\ell_2$ attacks. For standard or adversarially fine-tuned models, we use 25% of these budgets to enable a clear comparison.

## C.4 IMPLEMENTATION AND RESOURCES

**Implementation**. We utilize the Python programming language and PyTorch deep learning framework for our implementation (Paszke et al., 2019). Whenever possible, we utilize the default torchvision (TorchVision maintainers and contributors, 2016) implementations of our models - we modify these baselines for the changes mentioned above. For adversarial training and evaluation, we use the Adversarial Robustness Toolbox (Nicolae et al., 2018). Our source code provides further details regarding implementation[3].

**Hardware**. All experiments are conducted on the computational server of an institute, utilizing Nvidia L40S GPUs. The main paper experiments took a total of 600 GPU hours to complete, including $\geq 3$ seed replication for the main results. Total development time is estimated to be $3.5\times$ of the compute time for the final publication.

**LLM usage**. This work used LLMs to assist in literature search, phrasing, formatting, and implementation.

# D ADDITIONAL EMPIRICAL RESULTS

## D.1 EXPERIMENTS WITH OTHER DATASETS AND ARCHITECTURES

As mentioned in the main paper, we now extend our empirical findings to other datasets and architectures. Figure 13 demonstrates results with SVHN dataset and Wide ResNet 101-2 architecture (top), and CIFAR-100 dataset and VGG16 architecture (bottom). Our results replicate with novel datasets and architectures, as qualitatively identical results are obtained in these alternative settings. Note that the slight initial increase under neuron compressibility seen with WideResNet 101-2 here and ResNet18 in the main paper cannot be seen with VGG16, highlighting the regime dependence of multiple inductive biases compressibility-inducing regularizations might have.

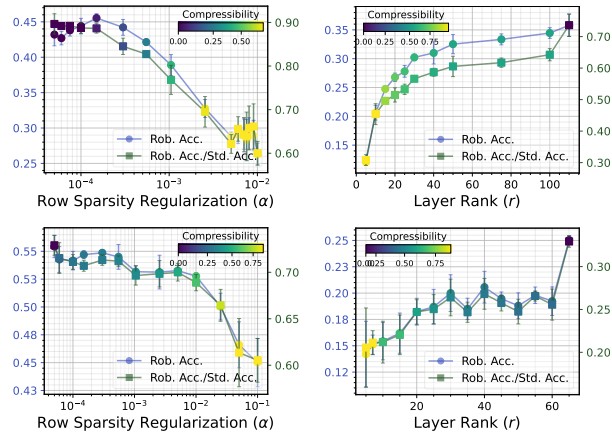

Figure 15: Results with CIFAR-10, FCN (top) and ResNet18 (bottom), with alternative attack norms.

## D.2 REGULARIZING GROUP SPARSITY

In the main paper, we highlight that we utilize a scale-invariant version of group lasso to disentangle the downstream effects of increasing compressibility vs. decreasing overall parameter scale. Figure 12 replicates our main results on ResNet18 and CIFAR-10 while using standard group lasso regularization. While its effects are mostly similar to our version of group lasso, we note that Figure 12 presents a subtle difference, where group lasso first creates an increase in robustness at very low levels (error bars = 1 std. deviation). However, as indicated in the main text, these benefits are overtaken by the negative effects of row compressibility as regularization strength increases.

---

[3]https://github.com/mbarsbey/advcomp

## D.3 ADVERSARIAL TRAINING RESULTS FOR SPECTRAL COMPRESSIBILITY

Figure 11 (left, center) presents the spectral compressibility counterpart for adversarial fine-tuning and training results from the main paper, under $\ell_2$ adversarial attacks. The patterns clearly mirror those presented in the main paper under row sparsity conditions.

## D.4 COMPRESSIBILITY THROUGH INDUCTIVE BIAS

We now examine whether the results we have observed with explicit regularization methods also apply to cases when compressibility is obtained through the inductive bias of the learning algorithm. For this, we go back to the setting presented in Appendix B, and instead of increasing regularization hyperparameter, we increase initial learning rate ($\eta$) of the training algorithm. The results, presented Figure 11 (right), paint an intriguing picture. While initially increasing $\eta$ *improves* adversarial robustness under $\ell_\infty$ attacks (perhaps paralleling its well-known benefits for standard generalization), as soon as it starts to increase row compressibility, its benefits of $\eta$ quickly disappear. This highlights the fact that our results not only inform the adversarial robustness behavior under explicit regularization and architecture design, but also inductive biases of the learning algorithm.

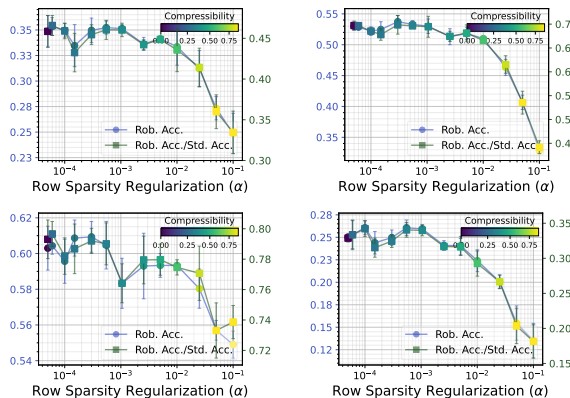

Figure 16: Results on CIFAR-10, ResNet18 and attacks with FGSM (top left), AutoCG (top right), Square Attack (bottom left), AutoAttack (bottom right).

## D.5 UNSTRUCTURED COMPRESSIBILITY

While unstructured compressibility is not the focus of our study, we note that it appears in the bound for $L_\Phi^\infty$ in Theorem 3.2, unlike that for $L_\Phi^2$. To investigate the significance of this result, we replicate the setting presented in Appendix B, but this time in addition to increasing the group lasso/nuclear norm regularization, we run a separate set of experiments where we either solely increase L1 regularization, or increase it along with structured sparsity-inducing regularization. We then compare the performance of the resulting models under the corresponding adversarial attacks. The results are presented in

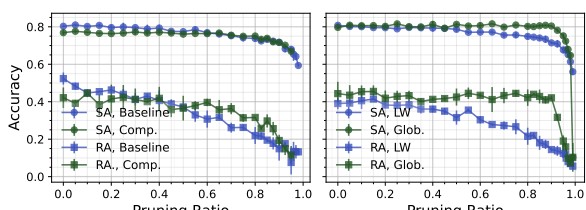

Figure 17: Post-pruning fine-tuning and robustness.

Figure 14. Remember that our bound implies *positive* effects of unstructured compressibility for $L_\Phi^\infty$. Indeed, in Figure 14 we see that L1 regularization can compensate for the negative effects of structured compressibility (top), while it has no such benefits for spectral compressibility (bottom). We believe that understanding the intricate relationships among different types of compressibility is a crucial future research direction.

## D.6 RESULTS WITH ALTERNATIVE NORMS, BUDGETS, AND ATTACKS

While for brevity we presented our main results to include robustness against $\ell_\infty$ attacks under neuron sparsity, and $\ell_2$ attacks under spectral compressibility, for completeness we provide our central results with the cross-norm attacks, *i.e.*$\ell_\infty$ attacks under spectral compressibility, and $\ell_2$ attacks under neuron sparsity. The results are presented in Figure 15, and are fully in line with the results presented in the main paper.

**Model performance under varying attack budgets**. As described in the main paper, in order to investigate the effects of structural interventions on standard trained models' adversarial robustness, we utilize a smaller attack budget to avoid floor effects from obscuring the effects we are investigating. Table 1 demonstrates that our results are not dependent on a specific attack budget, and the patterns that confirm our hypotheses hold across various attack budgets; however in standard trained models floor effects indeed prevent the observation of the results of our interventions, justifying our utilization of a reduced budget in such cases.

**Model performance under alternative attacks.**. We investigate whether our results replicate under alternative attacks. We therefore repeat our experiments with ResNet18 and CIFAR-10 in the main paper with FGSM (Goodfellow et al., 2015), AutoCG (Yamamura et al., 2022), Square Attack (Andriushchenko et al., 2020), and

Table 1: Robust accuracy of a ViT model trained on CIFAR-10, under increasing adversarial sample ratio in training ($\rho$) vs. increasing $\ell_\infty$ attack budgets ($\delta$).

|  | $\rho = 0.0$ | $\rho = 0.05$ | $\rho = 0.1$ | $\rho = 0.25$ | $\rho = 0.5$ |
|---|---|---|---|---|---|
| $\delta = 2/255$ | 0.111 | 0.333 | 0.479 | 0.519 | 0.510 |
| $\delta = 4/255$ | 0.002 | 0.061 | 0.263 | 0.371 | 0.390 |
| $\delta = 8/255$ | 0.000 | 0.002 | 0.032 | 0.113 | 0.179 |
| $\delta = 16/255$ | 0.000 | 0.000 | 0.000 | 0.005 | 0.019 |

the composite AutoAttack (Croce & Hein, 2020); as opposed to the original AutoPGD. Results in Figure 16 confirm that our results are qualitatively identical under different attacks.

## D.7 FINE-TUNING RESULTS WITH TRANSFORMERS

As described in the main text and above, we investigate whether we can replicate our results while fine-tuning ImageNet-pretrained transformer models, ViT and Swin Transformer, on CIFAR-10 and SVHN respectively, while utilizing sparsification regularization. The results are presented in Table 2 and Table 3, and replicate our hypotheses.

Table 2: Robust and standard accuracies of pretrained ViT models fine-tuned on CIFAR-10 dataset under varying neuron sparsification regularization strength ($\alpha$), i.e. group lasso.

|  | $\alpha = 0.0$ | $\alpha = 0.001$ | $\alpha = 0.005$ | $\alpha = 0.01$ | $\alpha = 0.05$ | $\alpha = 0.1$ |
|---|---|---|---|---|---|---|
| Rob. Acc. | 0.383 | 0.362 | 0.369 | 0.219 | 0.123 | 0.111 |
| Std. Acc. | 0.920 | 0.926 | 0.921 | 0.893 | 0.873 | 0.829 |
| RA/SA | 0.416 | 0.401 | 0.391 | 0.245 | 0.141 | 0.134 |

Table 3: Robust and standard accuracies of pretrained Swin Transformer models fine-tuned on SVHN dataset under varying neuron sparsification regularization strength ($\alpha$).

|  | $\alpha = 0.0$ | $\alpha = 0.001$ | $\alpha = 0.005$ | $\alpha = 0.01$ | $\alpha = 0.05$ | $\alpha = 0.1$ |
|---|---|---|---|---|---|---|
| Rob. Acc. | 0.384 | 0.360 | 0.357 | 0.326 | 0.155 | 0.083 |
| Std. Acc. | 0.889 | 0.877 | 0.887 | 0.880 | 0.881 | 0.875 |
| RA/SA | 0.432 | 0.410 | 0.402 | 0.370 | 0.176 | 0.095 |

Given that classification accuracy is the most commonly utilized and communicated metric in the literature on adversarial robustness, the main paper reports these as our primary metric. However, we find that same hypothesized patterns can be observed when robust loss - standard loss is utilized as the main metric, instead of accuracy. Table 4 demonstrates these results in the fine-tuning experiments described above, replicating our findings with robust and standard accuracy.

Table 4: Robust and standard accuracies and loss differences for pretrained Swin Transformer models fine-tuned on SVHN dataset under varying neuron sparsification regularization strength ($\alpha$).

|  | $\alpha = 0.0$ | $\alpha = 0.001$ | $\alpha = 0.005$ | $\alpha = 0.01$ | $\alpha = 0.05$ | $\alpha = 0.1$ |
|---|---|---|---|---|---|---|
| Rob. Acc. | 0.384 | 0.360 | 0.357 | 0.326 | 0.155 | 0.083 |
| Std. Acc. | 0.889 | 0.877 | 0.887 | 0.880 | 0.881 | 0.875 |
| Adv. Loss - Test Loss | 0.505 | 0.517 | 0.530 | 0.554 | 0.726 | 0.792 |

## D.8 RESULTS WITH POST-PRUNING FINE-TUNING

Utilizing a baseline model adversarially trained on CIFAR-10 dataset with ResNet18 architecture, instead of regularizing for compressibility, we prune and then fine tune our models to investigate 1- whether the main paper's results will replicate under post-pruning fine-tuning, 2- whether fine-tuning procedure will be another source of vulnerability in and of itself. After layerwise structured pruning, we fine-tune the models until convergence on the standard validation set. Our results, presented in Figure 17, demonstrate that 1- results from our main paper replicate under post-pruning fine-tuning, and 2- fine-tuning procedure creates an independent vulnerability - as after fine-tuning robustness deteriorates much faster compared to the standard accuracy vs. pre-fine-tuning results. Figure 18 demonstrates that the same results apply

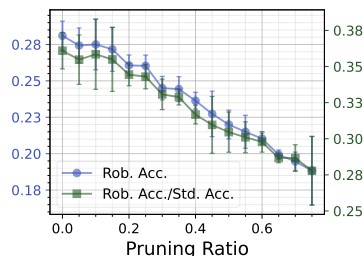

Figure 18: Adversarial post-pruning fine-tuning and robustness.

even when post-pruning fine-tuning is adversarial (conducted as defined above). These results are significant for both strengthening the main paper's conclusions, and for showcasing another compressibility-inducing intervention that leads to structure-induced vulnerabilities.

## D.9 EXPLOITATION OF VULNERABLE LATENT DIRECTIONS

Here we provide a more visual and intuitive walkthrough of our proposed mechanisms. Let us consider an MLP with a single hidden layer,

$$g(\boldsymbol{x}) = \mathbf{C}\phi(\mathbf{W}\boldsymbol{x}),$$

where $\phi$ corresponds to the elementwise ReLU function, and we ignore bias nodes without loss of generality for a cleaner exposition.

When two such networks have been trained on a dataset with no regularization vs. strong nuclear norm regularization, we can expect the latter's $\mathbf{W}$ to have much more concentrated singular values (SV), i.e. more spectrally compressible.

Indeed, in Figure 19, we provide a comparison of two such networks trained on CIFAR-10 (regularization strength $0$ vs. $0.05$), with hidden layer size $400$. We conduct a singular value decomposition (SVD) of $\mathbf{W} = \mathbf{U}\Sigma\mathbf{V}^\mathsf{T}$, and plot singular values of $\mathbf{W}$ for both networks $\boldsymbol{\sigma} := \mathrm{diag}(\Sigma) = (\sigma_1, \sigma_2, \dots)$. As in the main paper (Figure 2, left), in the compressible model the singular values are much more concentrated, creating the vulnerable directions in question.

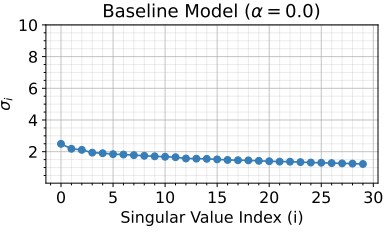

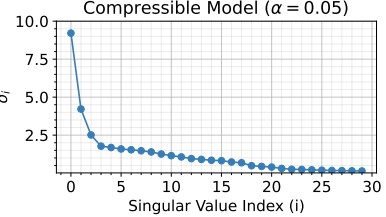

Figure 19: Comparing singular values of a baseline (top) vs. compressible (bottom) model.

But what exactly do we mean by attacks "aligning" with and "exploiting" these directions? For this, let us decompose an adversarial sample: $\boldsymbol{x}_{\mathrm{adv}} = \boldsymbol{x} + \boldsymbol{a}$, where $\boldsymbol{x}$ is the clean image and $\boldsymbol{a}$ is the adversarial perturbation. Examine the pre-activation representation of this attack:

$$\mathbf{W}\boldsymbol{x}_{\mathrm{adv}} = \mathbf{W}(\boldsymbol{x} + \boldsymbol{a}) = \mathbf{W}\boldsymbol{x} + \mathbf{W}\boldsymbol{a}.$$

Note that for a given sample $\boldsymbol{x}$, $\mathbf{W}\boldsymbol{x}$, and thus $\|\mathbf{W}\boldsymbol{x}\|_2$ are fixed. Having a large $\|\mathbf{W}\boldsymbol{a}\|_2$ (in relation to $\|\mathbf{W}\boldsymbol{x}\|_2$) would make it easier for the attacker to dominate the representation and change the downstream prediction.

So, how does the spikier $\boldsymbol{\sigma}$ in the compressible case help the adversary achieve this? For this, note that for every singular value $\sigma_i$, there exist the right and left singular vectors $\boldsymbol{u}_i$ and $\boldsymbol{v}_i$, constituting the columns of orthogonal matrices $\mathbf{U}$ and rows of $\mathbf{V}^\mathsf{T}$ respectively. So, based on the definition of SVD, we can write:

$$\mathbf{W}\boldsymbol{a} = \boldsymbol{u}_1\sigma_1\boldsymbol{v}_1^\mathsf{T}\boldsymbol{a} + \boldsymbol{u}_2\sigma_2\boldsymbol{v}_2^\mathsf{T}\boldsymbol{a} + \boldsymbol{u}_3\sigma_3\boldsymbol{v}_3^\mathsf{T}\boldsymbol{a} + \dots$$

Without loss of generality, let us assume $\|\boldsymbol{a}\|_2 = 1$, and examine these terms, $\boldsymbol{u}_i\sigma_i\boldsymbol{v}_i^\mathsf{T}\boldsymbol{a}$. Note that given both $\boldsymbol{v}_i$ and $\boldsymbol{a}$ are unit vectors, $\boldsymbol{v}_i^\mathsf{T}\boldsymbol{a}$ corresponds to *cosine similarity* of the two vectors, a very intuitive notion of alignment.

Why would $\boldsymbol{a}$ "align" with a $\boldsymbol{v}_1$ that has a large $\sigma_1$ (e.g. as in the leading SVs of the compressible model)? To see this, let us assume $\boldsymbol{a} \approx \boldsymbol{v}_1$. Then, this would mean $\boldsymbol{v}_1^\intercal \boldsymbol{a} \approx 1$, and $\boldsymbol{v}_j^\intercal \boldsymbol{a} \approx 0, \forall j > i$. This in turn would imply that

$$\|\mathbf{W}\boldsymbol{a}\|_2 = \|\boldsymbol{u}_1\sigma_1\boldsymbol{v}_1^\intercal\boldsymbol{a} + \boldsymbol{u}_2\sigma_2\boldsymbol{v}_2^\intercal\boldsymbol{a} + \boldsymbol{u}_3\sigma_3\boldsymbol{v}_3^\intercal\boldsymbol{a} + \dots\|_2$$
$$\approx \|\boldsymbol{u}_1\sigma_1 + 0 + 0 + \dots\|_2 = \|\boldsymbol{u}_1\sigma_1\| = \|\boldsymbol{u}_1\|\sigma_1$$
$$= \sigma_1$$

This means that after this layer $\boldsymbol{a}$ got scaled by this large number $\sigma_1$, helping it dominate the representation despite the small original attack budget:

$$\frac{\|\mathbf{W}\boldsymbol{a}\|}{\|\mathbf{W}\boldsymbol{x}\|} \gg \frac{\|\boldsymbol{a}\|}{\|\boldsymbol{x}\|}.$$

This example makes clear why having a few, very large $\sigma_i$ as a result of compression can create a large vulnerability. Note that Nern et al. (2023) also provide complementary theoretical justification regarding the dangers of encoders with such potent directions.

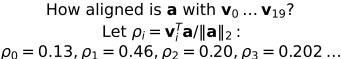

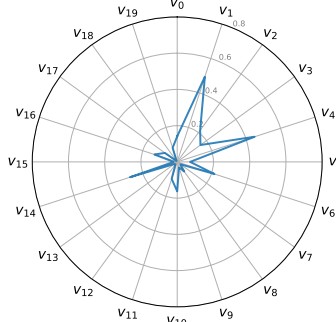

Figure 20: Examining alignment of a single adversarial perturbation with first 20 singular directions.

### D.9.1 ATTACK ALIGNMENT WITH LATENT DIRECTIONS

We now we move on to the question of whether this actually happens in practice. We already established that increased spectral compressibility creates vulnerable directions. How can we decide whether successful adversarial attacks are actually "exploiting" these directions? For any given $x$ and its perturbation $\boldsymbol{a}$, we can investigate the "alignment" of $\boldsymbol{a}$ with every singular direction $i$, we can compute $\rho_i = \boldsymbol{v}_i^\intercal \frac{\boldsymbol{a}}{\|\boldsymbol{a}\|_2}$, where we are now normalizing since $\boldsymbol{a}$ does not have to be unit norm in general. Note that $\rho_i \in [-1, 1]$ is a measure of alignment between $\boldsymbol{v}_i$ and $\boldsymbol{a}$; its absolute value

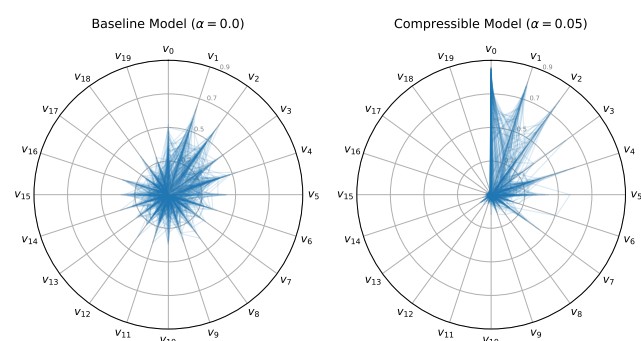

Figure 21: Comparing singular directions exploited by adversaries in baseline (left) vs. compressible (right) model.

$|\rho_i|$ can be utilized as a notion of *alignment strength*. An intuitive way to plot how much a sample aligns with each of the first $I$ singular directions is to plot this on a radar plot/spider plot. See Figure 20 for an example on a single $\boldsymbol{a}$. From this graph, we can read that $\boldsymbol{a}$ mostly aligns with $\boldsymbol{v}_1$, $\boldsymbol{v}_4$, and $\boldsymbol{v}_{14}$.

We can then use such a plot to understand overall patterns by plotting multiple samples. In Figure 21, we overlay this plot for 100 different samples for both models, for $I = 20$. The results *strongly support* our hypotheses: While the attacks in the baseline model exploit (i.e. align with) all 20 directions, in the compressible model the attacks focus on a few strong, *vulnerable* directions. Then, since the adversaries are using these potent directions at their

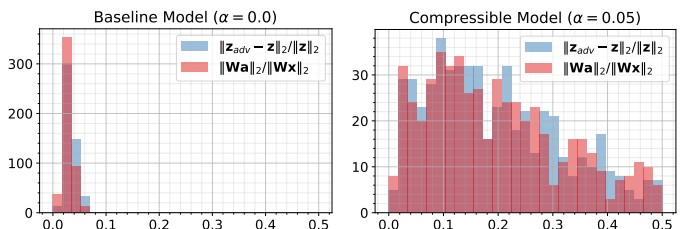

Figure 22: Comparing pre-activation ($\|\mathbf{W}\boldsymbol{a}\|_2/\|\mathbf{W}\boldsymbol{x}\|_2$) and post-activation ($\|\boldsymbol{z}_{\mathrm{adv}} - \boldsymbol{z}\|_2/\|\boldsymbol{z}\|_2$) representations of baseline (left) vs. compressible (right) models.

disposal in the compressible case, we would expect them to dominate the latent representations, compared to the baseline model. Indeed, in Figure 22, we see that this is indeed the case, both for pre-activation ($\|\mathbf{W}\boldsymbol{a}\|_2/\|\mathbf{W}\boldsymbol{x}\|_2$) and post-activation ($\|\boldsymbol{z}_{\mathrm{adv}} - \boldsymbol{z}\|_2/\|\boldsymbol{z}\|_2$) representations. Note that these results replicate the results presented in the main paper's Figure 4, right.

### D.9.2 WHITE BOX AND BLACK BOX ATTACKS EXPLOITING LATENT DIRECTIONS

We now can use this visualization technique to understand the *process of adversarial attacks finding these directions*. We choose two canonical, extensively cited white box and black box attacks for this task respectively: PGD (Madry et al., 2018) and NES (Ilyas et al., 2018).

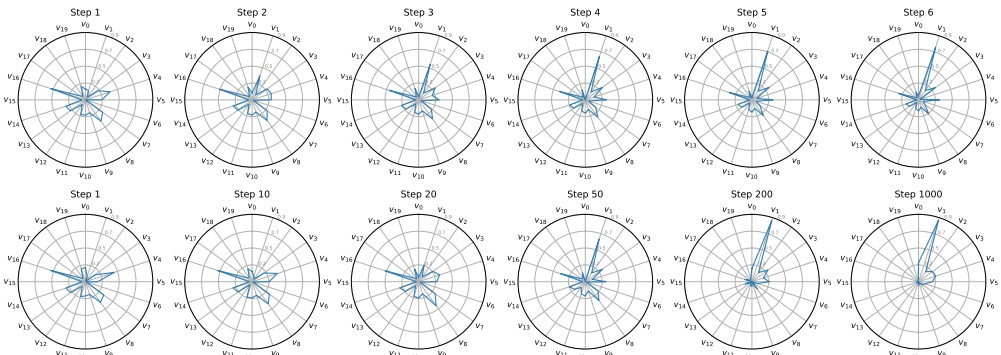

Figure 23: Utilization of singular directions by a white box PGD (top) vs. black box NES (bottom) attack under a compressible model.

As a white box attack that assumes access to parameters, PGD is able to conduct iterative first order optimization on the image to find a potent attack direction, projecting back to the $\epsilon$ ball after each iteration. Since in a compressible model, aligning with these sensitive directions would quickly increase the loss, the optimization algorithm very quickly finds these directions through its optimization objective. We present the iterates of a PGD perturbation on a single image under the compressible model, see Figure 23 (top). Note that the perturbation quickly aligns with a very strong singular direction, $\boldsymbol{v}_1$; so much so that by the 6th iteration, the algorithm converged on the attack already.

How can a black box attack make use of these exploitable directions? For this, let us take a closer look at NES. Being a black box attack, NES assumes only access to the logits, prohibiting the use of standard backpropagation. Instead, at every step, NES creates $N$ random Gaussian perturbations and evaluates the loss for all of them. It then calculates a weighted average of these directions (weighted by their impact on the loss) to estimate a *proxy gradient*, and update the adversarial perturbation accordingly. This means that whenever these random perturbations align, even slightly, with any of the exploitable latent directions, they would dominate the weighted average, effectively pulling the estimated gradient toward the vulnerability. So, although not as efficiently, without an explicit knowledge of the parameter space, NES can locate these *adversarially exploitable directions*, just by querying the input space. Indeed, the image in Figure 23 (bottom) presents a direct confirmation of this hypothesis: although it takes many more steps ($\sim 200$) due to the randomness of its perturbations, NES can also converge to exploiting the vulnerable directions in the latent space.

Note that we focused on the $\ell_2$ attacks in this exposition; however note that it is quite straightforward to apply a similar analysis to $\ell_\infty$ case, where the most vulnerable directions are rows $\mathbf{w}_k$ in $\mathbf{W}$ with the largest $\|\ell_k\|_1$.

### D.9.3 INTERLAYER ALIGNMENT

Following from previous example, let us now assume a two layer neural network $g(x) = \mathbf{C}\phi(\mathbf{W}^1\phi(\mathbf{W}^0))$ - we will use superscripts to denote the components of layers as well. As a simple example, assume that both layers have a single, very large SV, and rest of their SVs are $\approx 0$. This implies that both have a potent singular direction that can potentially be exploited. However, these directions between layers will need to "align" for their impact to accumulate. More concretely, note that a unit perturbation $\boldsymbol{a}$ that aligns with the right singular vector of the first layer $\mathbf{W}^0$ in the input space $\boldsymbol{a} \approx \boldsymbol{v}_1^0$ will be "amplified" by $\sigma_1^0$. The resulting output will be in the direction of the left singular vector, i.e. $\boldsymbol{u}_1^0\sigma_1^0$. Ignoring nonlinearity for now, as large as this intermediate representation can be, if it's not in the direction of the next layers' large SV, it will be effectively ignored. For example, at the extreme end, if $(\boldsymbol{v}_1^1)^T\boldsymbol{u}_1^0 \approx 0$, the attack will effectively disappear before it reaches the final representation and can impact the prediction. This is because in the second layer we will have $\|\mathbf{W}^1\boldsymbol{u}_1^0\|_2 \approx 0$. So, such a theory will have to take into account how such signals are relayed between layers, while factoring in nonlinearity.

Note that Theorem 3.2 upper bounds $L_\Phi^p$, the $p$-norm Lipschitz constant of the encoder. This can be computed as the maximum $p \to p$ operator norm of the Jacobian:

$$L_\Phi^p = \sup_{\boldsymbol{x}\in\mathcal{X}} \|\mathbf{J}_\Phi(\boldsymbol{x})\|_p = \sup_{\boldsymbol{x}\in\mathcal{X}} \|\mathbf{D}^\lambda(\boldsymbol{x})\mathbf{W}^\lambda\mathbf{D}^{\lambda-1}(\boldsymbol{x})\mathbf{W}^{\lambda-1}\dots\mathbf{D}^1(\boldsymbol{x})\mathbf{W}^1\|_p, \qquad (79)$$

where the diagonal binary $\mathbf{D}^l(\boldsymbol{x})$ terms stand for the ReLU nonlinearity. Notice that input dependence of these terms introduce a combinatorial complexity, making it infeasible to directly optimize this term. Our Theorem 3.2, like all other attempts in the literature, utilizes an approximation of this monolith.

Given that $\|D^l\|_p = 1$, and submultiplicativity of the operator norm, it is possible to write:

$$L_\Phi^p \leq \sup_{\boldsymbol{x} \in \mathcal{X}} \|\mathbf{D}^\lambda(\boldsymbol{x})\mathbf{W}^\lambda \mathbf{D}^{\lambda-1}(\boldsymbol{x})\mathbf{W}^{\lambda-1} \ldots \mathbf{D}^1(\boldsymbol{x})\mathbf{W}^1\|_p, \tag{80}$$

$$\leq \|\mathbf{W}^\lambda\|_p\|\mathbf{W}^{\lambda-1}\|_p \ldots \|\mathbf{W}^1\|_p. \tag{81}$$

While this is valid, notice that it corresponds to a very pessimistic assumption: It looks at how much every layer can maximally "stretch" an incoming vector, and multiplies this across layers. This assumes that all "worst case" directions in consecutive layers exactly line up.

Instead, in our bound, while layer operator norms appear (through their compressibility-based decomposition), interlayer alignment terms, $A_p(l) \leq 1$, act as a *correction term*.

$$L_\Phi^p \leq \|\mathbf{W}^\lambda\|_p A_p(\lambda - 1)\|\mathbf{W}^{\lambda-1}\|_p A_p(\lambda - 2) \ldots A_p(1)\|\mathbf{W}^1\|_p. \tag{82}$$

It approximates and factors in how much *dominant directions* actually align in consecutive layers. Every $A_p(l)$ consists of two terms: the main term that computes the alignment of the dominant directions, and a remainder term that goes to 0 as compressibility increases. See Appendix B.5 for how we approximate this (much more manageable) combinatorial computation. We refer the reader to our proofs for a full derivation of these terms. Note that while this term is not the main focus of our paper, Figure 10 includes empirical investigation of this term, and demonstrates that it does not have a strong directional relationship with compressibility.

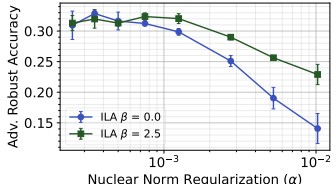

Figure 24: Effects of regularizing interlayer alignment (ILA).

**Utilizing interlayer alignment for regularization**. In order to provide a more comprehensive examination of this term, we conduct experiments that test whether this term can be used as another theoretically inspired *intervention for robust compressibility*. With a linear approximation to this term (regularizing $\|(\mathbf{U}_k^l)^T \mathbf{V}_k^{l+1}\|_F^2$), we test this hypothesis. Results, presented in Figure 24, are directly in line with our predictions: regularizing interlayer alignment between layers lead to a tangible increase in robustness under compressibility. While some mild computational hurdles need to be addressed for full practical utility, these results both provide a new intriguing research direction for robust compression, as well as serving as a yet another confirmation of our theory.

### D.10    COMPARISON WITH ADVERSARIAL PRUNING LITERATURE

As discussed in the main paper, we consider our work to be complementary to those in the field of adversarial pruning (Piras et al., 2025). More specifically, our theory implies that compressibility hurts robustness insofar as it increases operator norms and creates adversarially vulnerable directions in the latent space; we thus investigate two structured adversarial pruning methods from the literature to see whether these successful adversarial pruning methods implicitly control operator norms. For this, we investigate HARP (Zhao & Wressnegger, 2023) and grouped kernel pruning (GKP) (Zhong et al., 2023). We choose these two as they have distinct motivating hypotheses, neither of which is in common with ours in a meaningful way. We use both papers' official repositories to conduct adversarial pruning with a ResNet18 on CIFAR-10. As baselines, for HARP we train a uniform/layerwise pruning algorithm with standard training set, for GKP we replace grouped kernel pruning with standard filter pruning.

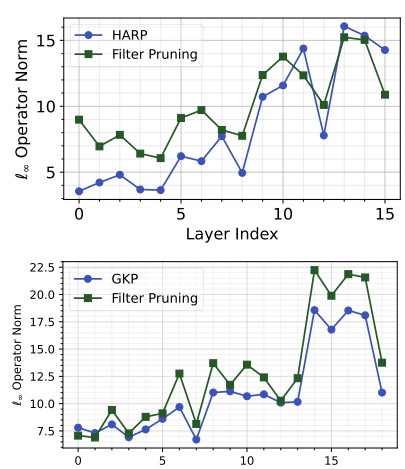

Figure 25: Operator norms of models under adversarial pruning vs. baselines.

After the training, we measure the $\ell_\infty$ operator norms for both methods and compare it to their baselines. Intriguingly, as shown in Figure 25, although neither method conducts operator norm control *explicitly*, we find that both end up controlling operator norms *indirectly*. Note that this cannot just be a by-product of adversarial training as GKP relies solely on filter restructuring, and does not involve any adversarial training. We find this to be a promising first finding towards a comprehensive understanding of robust structured compression.

### D.11    FURTHER EXPERIMENTS WITH UAEs

We first replicate our original results with a ResNet18 trained on CIFAR-10 in Figure 26. Note that the $x$-axis in this particular figure represents the fixing of the Frobenius norms to $x$-times their initialization norms - this allows us to fix norms using a common value for layers that have widely different widths (while for FCN we used a single constant). Our results qualitatively replicate those in the main paper.

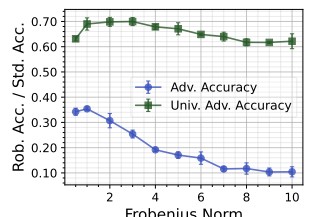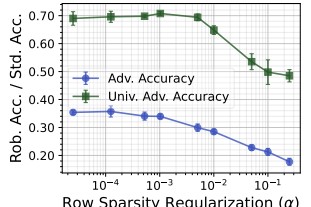

Figure 26: Robustness against standard vs. universal adversarial attacks under changing Frobenius norm coefficient (left) vs. group lasso (right).

To further probe this causal relationship, we conduct adversarial training with ResNet18s under increasing compressibility. Importantly, we conduct the training either with standard adversarial examples vs. UAEs. Given the computational challenges of computing UAEs at every iteration, we use the cheaper FGSM attack for universal and standard adversarial samples, generated from 0.1 of the input batch. Results, presented in Figure 27, illustrate the average spread ($\beta$) of the dominant terms in the networks under UAE vs. standard adversarial training. Our findings show that UAE training dramatically reduces spread of the dominant terms compared to standard

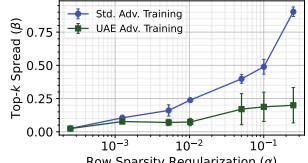

Figure 27: $\beta$ and UAEs.

adversarial training, implying that just as creation of vulnerable latent directions allow UAEs, training against them reduces the potency of such directions, providing convergent evidence for our hypotheses.

