# OpenReview forum: "On the Interaction of Compressibility and Adversarial Robustness"
_ICLR.cc/2026/Conference — ICLR 2026 Poster_

### Official Review · Reviewer_1FKt · 2025-10-25

**Soundness:** 2
**Presentation:** 3
**Contribution:** 3
**Rating:** 6
**Confidence:** 3

**Summary:**

To understand the impact of model compression on adversarial robustness, this paper presents an adversarial robustness bound that interprets how structured and spectral compression induce adversarial vulnerability through their effects on the model's Lipschitz constant. Extensive experiments further demonstrate the detrimental impact of model compression on adversarial robustness.

**Strengths:**

**Valuable insight into model compressibility in terms of adversarial robustness**

This work provides a unified theoretical analysis that helps to clarify the relationship between model compressibility and adversarial vulnerability without being constrained to a specific norm-based perturbation.

**Extensive experiments across various settings**

This paper presents extensive experiments across different model architectures, learning mechanisms, and adversarial example generation methods, offering a thorough and comprehensive analysis of the relationship between structured compressibility and adversarial robustness.

**Weaknesses:**

**W1: The scenario of generating model compression is unclear.**

Model compression can be achieved either through a fine-tuning procedure using the full or partial training dataset, or in a data-free manner. In particular, for the former case, model compressibility is closely related to the feature representations of the training dataset in terms of model capacity. I would assume that the considered compressibility is restricted to a fine-tuning procedure; however, a clearer introduction to the model compression setup is needed before the discussion of compressibility.

**W2: Representation of sensitivity along a small number of directions.**

The authors claim that structured compression induces high sensitivity to adversarial perturbations along a small number of directions. However, this conclusion cannot be directly drawn merely from the observed correlation between compressibility and adversarial robustness. Lines 216–223 explain that the potent attack directions are determined by interlayer alignment; however, these directions are neither visualized nor formally defined.

**W3: Unclear explanation for the amplification of adversarial attacks in the representation space.**

The authors attribute the high adversarial vulnerability of compressed models to amplification along certain sensitive directions. However, it remains unclear how the adversarial attack is amplified and what mechanisms cause this amplification to occur.

**W4: Lack of analysis on robustness-aware model pruning techniques.**

This work discusses compressibility mainly after model training. However, many prior approaches achieve better preservation of both standard accuracy and adversarial robustness simultaneously during structured model pruning [1,2,3]. Despite the observed proportional relationship between compressibility and adversarial vulnerability, an additional analysis is needed to investigate the impact of adversarially robust model pruning.

---
### References

[1] Zhao and Wressnegger, "Holistic Adversarially Robust Pruning", ICLR 2023.
[2] Sehweg et al., "Hydra: Pruning Adversarially Robust Neural Networks," NeurIPS 2020.
[3] Ye et al., "Adversarial robustness vs. model compression, or both?" ICCV 2019.

**Questions:**

**Q1: The rationale of interlayer alignment**

I acknowledge the importance of the alignment between consecutive layers. However, in Lines 216–236, it is not clear what *interlayer alignment* exactly refers to. Moreover, I am curious about the rationale behind the formal definitions of $A^{\ast}_{\inf}$ and $A^{\ast}_2$. How do these two terms represent interlayer alignment?

**Q2: The impact on robustness improvement with small compressibility**

For CNNs (Figure 5) and Transformers (Figure 6), a relatively small spectral compressibility can result in a slight improvement in adversarial robustness. How can this positive effect be explained according to the proposed theory?

---

> ### Author Response · Authors · 2025-11-25
>
> We thank the reviewer for their insightful comments and feedback, as well as positive reception of our work. We hope that our responses below address their concerns, leading to an upwards revision in their score. We remain available to address further concerns.
>
> ---
>
> **W1: Compression scenarios.** We are primarily interested in the robustness effects of **interventions that facilitate structured compression**, including explicit regularizations, architectural design, or implicit bias of the optimization. To retain equivalence between various interventions we experiment with, we **compare models before conducting compression**, while explicitly visualizing their sparsity/compressibility. This prevents the particulars of the compression methodology to confound the effects of structured compressibility. We then extend our results to **realistic compression scenarios** to ensure **practical relevance**. Note that e.g. for low-rank factorization there is no separate compression procedure. On the other hand, for an intervention like row sparsification (group lasso), after demonstrating our results in uncompressed models, we demonstrate that our conclusions **remain identical after filter pruning** - including before (Section 5) or after (Appendix D.8) fine-tuning. We now extend the discussion of this in the main paper **(L318-323)** and show that even basic post-pruning fine-tuning can create such vulnerable directions and increase operator norms **without any additional regularization** (**L449-453**; Appendix D.8), further highlighting the practical relevance of our results.
>
> ---
> **W2/W3: Mechanism of compressibility-induced adversarial attack amplification.** We would like to highlight that our conclusions are **not based on observations of correlations**, but is directly implied by our theory and systematically confirmed by our experiments. However, we appreciate the invitation to provide **stronger intuition** regarding how compressibility *actually* enables adversaries. We introduced a **new section in the Appendix (D.9)** that takes the reader through the implications of our theory *step-by-step*, with more detailed exposition, intuitive examples, and visualizations - including how exactly **white box _and_ black box** attacks can exploit these directions, as well as a detailed discussion of **interlayer alignment**. To avoid repetition, we kindly refer the reviewer to our response to Reviewer TtqN (W1/Q1) for an abridged account of how adversaries can exploit vulnerable directions for greater representation dominance, and continue to discussing **interlayer alignment**.
>
> ---
> **W2/Q1: Interlayer alignment.** Let us assume a network with two hidden layers $g(x) = \mathbf{C}\phi(\mathbf{W}^1\phi(\mathbf{W}^0))$ with each layer parameter having single, very large non-zero singular value ($\sigma^1_1$ and $\sigma^0_1$). This implies that both have potent singular directions that can be exploited, however, how much a representation will be amplified is also **dependent on how these layers' directions "align"** for their impact to accumulate. More concretely, note that an adversarial perturbation $\mathbf{a}$ that aligns with the right singular vector of the first layer $\mathbf{a} \approx \mathbf{v}^0_1$ will be "amplified" by $\sigma^0_1$. The resulting output will be in the direction of the left singular vector, i.e. $\mathbf{u}^0_1\sigma^0_1$. Ignoring nonlinearity for now, if this direction is *orthogonal* to next layer's $\mathbf{v}^1_1$, i.e. $(\mathbf{v}^1_1)^T \mathbf{u}^0_1 \approx 0$, the attack will effectively be "quashed" before it reaches the final representation and can impact the prediction, since $||\mathbf{W}^1\mathbf{u}^0_1||\_2 \approx 0$. So, such a theory will have to take into account how **such signals are relayed between layers**, while factoring in nonlinearity (See Appendix D.9 for a more detailed narrative).
>
> ---
> ### Continued below

---

> > ### Author Response · Authors · 2025-11-25
> >
> > ### Continued from above
> > ---
> >
> >
> > Note that **Theorem 3.2** upper bounds $L^{p}\_{\Phi}$, the $p$-norm Lipschitz constant of the encoder. This can be computed as:
> > $$
> >     L^{p}\_{\Phi} = \sup\_{\mathbf{x}\in \mathcal{X}} ||\mathbf{D}^{\lambda}(\mathbf{x})\mathbf{W}^{\lambda} \mathbf{D}^{\lambda-1}(\mathbf{x}) \mathbf{W}^{\lambda-1} \dots \mathbf{D}^{1}(\mathbf{x}) \mathbf{W}^{1}||\_{p},
> > $$
> > where the diagonal binary $\mathbf{D}^{l}(\mathbf{x})$ terms stand for the **ReLU**. Notice the **combinatorial complexity** of RHS, making it **infeasible** for direct optimization. Since $||\mathbf{D}^{l}||\_p = 1$, using submultiplicativity of the operator norm, it is valid to write:
> > $$
> >     L^{p}\_{\Phi} \leq ||\mathbf{W}^{\lambda}||\_p ||\mathbf{W}^{\lambda-1}||\_p \dots ||\mathbf{W}^{1}||\_{p}.
> > $$
> > However, this corresponds to a **very pessimistic assumption**: that all "worst case" directions in consecutive layers **exactly line up**. Instead, our bound introduces **interlayer alignment** terms, $A\_p(l) \leq 1$,  act as a **correction term** to this product:
> > $$
> >     L^{p}\_{\Phi} \leq ||\mathbf{W}^{\lambda}||\_p A_p(\lambda-1) ||\mathbf{W}^{\lambda-1}||\_p A\_p(\lambda-2) \dots A\_p(1)||\mathbf{W}^{1}||\_{p}.
> > $$
> > It approximates and factors in how much dominant directions actually align in consecutive layers. Every $A_p(l)$ consists of two terms: the main term that computes the **alignment of the top-$k$ dominant directions**, and a remainder term that goes to $0$ as compressibility increases. We revised this section to increase accessibility of this term with a more **streamlined notation** and **stronger intuition** **(L228-248)**, and provided a more detailed version of this account in Appendix D.9.
> >
> > To further examine this term, we test whether this term can be used as another **theoretically-informed regularization for robust compressibility**. Results, presented in [Figure 24](https://anonymous.4open.science/api/repo/11954/file/figure_24_interlayer.png?v=2f316180) as well as in the new Appendix D.9.3, **directly confirm** our predictions: regularizing $A_p(l)$ leads to a tangible increase in robustness under compressibility. We consider this a new intriguing **research direction** for robust compression, as well as another **confirmation** of our theory. We highlight these results in the revised paper as well **(L462-466)**.
> >
> > ---
> > **W4: Adversarial pruning methods.** We see our work as being **complementary to adversarial pruning (AP)** literature (Piras et al., 2025). Such methods are usually characterized by multi-objective optimization procedures for robust accuracy and compression. We provide a theoretically informed complementary perspective for such future procedures, confirmed by our demonstration that **all major terms** that appear in our bound can be utilized for robust compression regularization. To pursue our findings' complementarity with this literature further, we investigate two structured AP methods from the literature. Our theory implies that compressibility hurts robustness **insofar as it increases operator norms** and creates adversarially vulnerable directions; so we ask if we can observe successful adversarial pruning methods **implicitly control operator norms**. For this, we investigate HARP [1] and GKP (Zhong et al. 2023), two methods with **distinct motivating hypotheses** from each other *and* us. As baselines, for HARP we train a uniform pruning algorithm with standard training, and for GKP we replace grouped kernel pruning with standard filter pruning.
> >
> > After the training, we measure the $\ell_\infty$ **operator norms** for both methods and compare it to baselines. Intriguingly, as shown in [Figure 25](https://anonymous.4open.science/api/repo/11954/file/figure_25_adv_pruning.png?v=dfb112de), we find that both methods **implicitly control operator norms**; providing a *complementary explanation* for their success from our perspective. Note that this is **not a by-product of adversarial training** as GKP does **not** utilize adversarial training. We find this to be an **exciting first finding** towards a comprehensive understanding of robust structured compression. We incorporated this discussion into the main paper and directed the reader to these findings in Appendix D.10 **(L513-518)**.
> >
> > ---
> > **Q2: Benefits of limited compressibility.** This **sharp observation** is compatible with our theory,  as our theory pertains to the regime where structured sparsification becomes **strong enough** to create few dominant latent directions (as noted in L467-569). At lower levels, such interventions can act as **beneficial capacity control**. However, this effect appears context-dependent; for instance, our VGG16 experiments (Figure 13, bottom left) do not show this initial benefit. We have updated this Appendix section to discuss these regime-dependent trade-offs **(L1499-1502)**.

---

### Official Review · Reviewer_LZxs · 2025-10-28

**Soundness:** 4
**Presentation:** 3
**Contribution:** 2
**Rating:** 4
**Confidence:** 4

**Summary:**

This paper proposes a systematic theoretical framework for analyzing how structured compressibility — specifically,  neuron-level compressibility and spectral compressibility, affect model’s adversarial robustness. The authors propose to characterize l∞ and l2 operator norms of the parameters by an upper bound that decomposes into (compressibility × Frobenius norm) terms. Building on this formulation, they further derive and analyze an upper bound on the network’s overall Lipschitz constant. They show that compression introduces a few highly sensitive directions that can significantly amplify perturbations, which can then be exploited by attackers, ultimately leading to degraded robustness. The experimental section covers a wide range of architectures, such as FCNs, CNNs and Transformers, validating the theoretical predictions that models with higher structured compressibility are more vulnerable to adversarial perturbations. The results also demonstrate that the vulnerabilities induced by compression persists even under adversarial training and transfer learning, and facilitates the emergence of universal adversarial perturbations.

**Strengths:**

1. Provides a unified norm-based framework connecting structured compressibility and adversarial robustness.
2. Characterizing the l∞ and l2 operator norms of the parameters by decomposing the effects into compressibility and Frobenius norm terms, thereby further formalizing an upper bound on the model’s Lipschitz constant.
3. The analysis shows that the impact of compressibility on robustness persists in adversarial training and transfer learning, and it can facilitate the emergence of universal adversarial perturbations.
4. The theoretical analysis, mathematical derivation, and experimental process are relatively complete.

**Weaknesses:**

Although the theoretical analysis is mathematically sound and logically consistent with prior robustness theory, the overall reasoning builds on well-known intuitions (ideas such as model structured compression concentrates sensitivity along a small number of directions in representation space, which in turn results in decreased robustness), and the upper bound mainly formalize this intuition rather than uncover new mechanisms. The experiments, though thorough, largely confirm expected behaviors without surprising counterexamples or deeper causal probes. The theoretical results appear to just extensions, restating known connections in a more formalized way. In addition, the two interventions mentioned in the paper for improving robustness also seem to have been studied.

**Questions:**

1. The main theoretical results seem to formalize known connections between compressibility, operator norms, and robustness. Could the authors further clarify the new theoretical insights or perspectives provided by their analysis beyond existing results?
2. While the paper formalizes an upper bound on adversarial robustness in terms of compressibility and Frobenius norm, the bound appears relatively loose—being an order of magnitude above the empirical robustness gap as the authors describe in appendix. As such, its practical utility for predicting or guiding robust model design seems limited.
3. The theoretical analysis appears focused on ℓ₂ and ℓ∞ perturbations. Does the same framework extend to other robustness notions (e.g., ℓ₁, distributional robustness, or label noise)?

---

> ### Author Response · Authors · 2025-11-25
>
> We thank the reviewer for their feedback and acknowledgment of the strengths of our work. We hope that our comments below, together with the revisions made to the paper and the new supporting results, will address their concerns and lead to a positive reconsideration of our work.
>
> ---
> ### W1/Q1: Uniqueness, novelty, and significance of our contributions
>
> The reviewer's main concern is that our contribution merely formalizes known intuitions with limited novel insight, **a premise we must respectfully disagree with**. Given the current state of the literature, we find this assessment *unexpected*: to our knowledge, there is **no consensus** even on the **overall direction** of the relationship between compressibility and robustness, let alone on the _specific mechanisms_ we propose. Therefore, in this rebuttal we  focus on clarifying the context of our work and highlighting its **unique contributions**; we have also incorporated these clarifications into the revised paper. If the reviewer has specific studies in mind, we would be happy to discuss how they **relate to our framework**.
>
> The existing literature does not present a **unified view** of the relationship between compressibility/sparsity and adversarial robustness (AR): there are works that argue for a **positive or negative** relationship (as summarized in L495-498). More specifically for structured compressibility (SC), there is almost **no theoretical guidance**. To check whether the *specific mechanisms* we study appear in practice as informal motivations/intuitions, we examined a recent review of adversarial pruning methods (Piras et al., 2025) and the structured pruning techniques cited therein. Among the *13 methods* cited, we found **no evidence** of methods explicitly motivated by compressibility-induced vulnerable latent directions, dominance vs. spread of leading terms, or interlayer alignment as we define it. We also note that _none_ of them are **theoretically motivated**. To our knowledge, the works closest to ours are Savostianova et al. (2023) and Feng et al. (2025), which study how increasing condition numbers due to unstructured sparsity and low-rank training, respectively, affect robustness. We use these as **primary comparison points** below.
>
> ---
> ### Q1: Novel Insights from Our Framework
>
> We now highlight the **main novel insights** produced by our framework and the experiments that confirm them.
>
> **Source-agnostic characterization of SC-AR relationship.** To our knowledge, we provide the **first comprehensive characterization** of the SC-AR relationship, and our theory is explicitly **source-agnostic** (Theorem 3.2): it applies (and is empirically confirmed) *regardless* of whether compressibility arises from explicit regularization (L335), architectural design (L336), optimization-based inductive bias (Appendix D.4), or post-pruning fine-tuning (Appendix D.8). Similarly, we show that the effect of compressibility **persists under adversarial training** (Figure 7, left). In contrast, for example, Feng et al. (2025) analyze robustness *only* under the increasing unstructured sparsity, and Savostianova et al. (2023) *only* under low-rank training. Neither study investigates **neuron compressibility** at all.  We incorporated this important comparison to the Section 5 of our paper **(L506-513)**.
>
> ---
>
> **Structure vs. scale distinction.** A key contribution of our theory is the explicit **structure vs. scale decomposition** of operator norms (Theorem 3.2), which *disentangles* compressibility from *overall parameter scale* (Frobenius norm). This allows us to reason about **familiar training interventions** through this lens. For example, our results show that reducing the Frobenius norm (e.g., via weight decay) is beneficial for robustness, whereas achieving a similar reduction via regularizers that also induce *structural imbalance* (e.g., nuclear norm regularization) has more ambiguous and ultimately negative effects (e.g. Figure 12 visualizes this “*tug-of-war*’’ between these two components). This **decomposition** also leads to **new predictions**: in Figure 7 (center left, center right), we confirm that while both structured compressibility and Frobenius norm increase adversarial vulnerability, only the former increases **vulnerability to UAEs**. To our knowledge, this asymmetric behavior with respect to UAEs has **not been observed or predicted before**. In new results (see our response to Reviewer Tszb and the new Appendix D.11), we replicate these results and also show that adversarial training with UAEs, but *not* with standard adversarial examples, **reduces the spread of the top-$k$ terms** ($\beta$), supporting our hypotheses in the *converse direction*. We incorporated these results into the revised paper **(L422-425)**.
>
> ---
> ### Continued below

---

> ### Author Response · Authors · 2025-11-25
>
> ### Continued from above
> ---
>
> **Dominance vs. spread.** Unlike prior work that relies primarily on condition number (see above), our framework uses a more fine-grained notion of compressibility that explicitly distinguishes **dominance vs. spread** of the leading $k$ terms. This implies that two networks with *equally dominant* leading terms (same $k, \epsilon$) can nevertheless have **very different safety profiles** depending on the **spread** of those terms ($\beta$), and suggests concrete *design principles* for interventions (Section 4.1). We provide a formal comparison with alternative sparsity/compressibility measures in Appendix B.1, and show that our notion better distinguishes **qualitatively different parameter configurations**. We highlighted this point in our text **(L151-153)**.
>
> ---
> **Robustness of pretrained encoders.** While many Lipschitz-based analyses focus on the Lipschitz constant of the loss, we also analyze it with respect to the learned representations (as well as the loss; see Corollary 3.3). This lets us predict and confirm that **compressibility-induced vulnerability transfers** to downstream tasks after fine-tuning (Figure 7, right), further confirmed by our **zero-shot classification CLIP** experiments (Figure 6, right).
>
> ---
> **Theory-based interventions for robust compression.** Our interventions are designed both to **stress-test our theory** and to **inform future robust compression strategies**. In addition to terms directly related to compressibility, we show in *new experiments* (see our response to Reviewer 1FKt) that **regularizing interlayer alignment**, another quantity in our bound that pertains to dominant terms, yields **tangible robustness gains**. Combined with our main experiments, this demonstrates that **each major term** of Theorem 3.2 can be targeted to improve robustness, validating our theory and providing theory-informed guidance to future methods. We expanded our Section 4 accordingly **(L461-466)**. Although known regularizers such as operator norm or condition-number regularization *might* have similar, unintended effects, we are not aware of any existing methods that **directly** target top-$k$ spread or interlayer alignment in  for robust structured compression. Similarly, for example, although Diao et al. (2023) use inherent (*unstructured*) parameter sparsity to conduct global pruning, they provide theoretical results that *only* connect pruning ratio to their proposed sparsity index, **not to robustness performance**.
>
> ---
> **Summary: A safety-critical scenario.** By way of summary, we assess a **practically relevant**, **safety-critical** scenario. Suppose an encoder (e.g., CLIP, DINO) is trained or fine-tuned under strong structured sparsification, as may occur due to **real-time inference constraints**. Our theory and experiments jointly imply that: (1) such compressibility **creates adversarial vulnerability regardless of its source**; (2) this vulnerability **transfers to downstream tasks** (e.g., autonomous driving); and (3) this setup **facilitates UAEs**, i.e., a successful perturbation can be **reused** across many inputs. Each component of this scenario is both *predicted* by our theory and empirically *validated* in our experiments. We are not aware of any previous framework that can **cohesively explain and predict all these aspects**.
>
> ---
> ### Q2/Q3 Bound tightness, alternative robustness metrics
>
> **Q2: Bound tightness.** Our bounds are intended as **analytical tools** to understand how structured compressibility-related variables affect adversarial risk and network Lipschitz constants, rather than as tight estimators of empirical robustness gaps. This is consistent with many **existing norm-based bounds** (Arora et al., 2018; Wen et al., 2020), whose value lies in **disentangling** the roles of different factors while maintaining correlation with empirical behavior, as we do (Figure 3). Moreover, the experiments throughout Section 4 repeatedly **test and confirm our theoretical predictions**, testifying to its ability to help guide robust compression. We are not aware of any previous work that provides tight Lipschitz bounds in relation to structured compressibility. However, combining our approach with more refined Lipschitz estimators (e.g., ECLipsE by Xu et al., 2024) is indeed a **promising direction**. We now highlight this explicitly in our paper **(L534-536)**.
>
> ---
>
> **Q3: Alternative robustness notions.** Given the limited prior work in this area, we focused on the most commonly used attack types, in line with previous studies. Our approach can in principle be extended to any $p$-$q$ dual norm pair; for example, a direct extension to $\ell_1$ attacks would relate them to column/input feature map compressibility. We view extending our framework to **other attack norms**, as well as broader notions of **out-of-distribution robustness** to be compelling future directions that we now mention explicitly in our paper **(L535-538)**.

---

> > ### Comment · Reviewer_LZxs · 2025-11-25
> > **Acknowledgment of Rebuttal**
> >
> > I appreciate the detailed rebuttal and the clarifications you have provided. Please note that I am currently reviewing your responses carefully and will update my assessment accordingly. I kindly ask for your patience while I complete this process.

---

> > > ### Author Response · Authors · 2025-11-26
> > >
> > > We thank the reviewer again for their time and attention. We remain fully available throughout the discussion phase.

---

### Official Review · Reviewer_Tszb · 2025-10-31

**Soundness:** 3
**Presentation:** 4
**Contribution:** 3
**Rating:** 6
**Confidence:** 4

**Summary:**

The paper investigates the fundamental relationship between network compressibility and adversarial robustness. The claim is that their interaction remains poorly understood. They show theoritical bounds and also empirical evaluation on architectures (FCNs, CNNs, VIT) and multiple datasets. Results show that Increased neuron or spectral compressibility consistently reduces adversarial robustness, even under adversarial training.

**Strengths:**

- Paper is well motivated and well written
- Provides a well-explained theoretical contribution between compressibility and adversarial robustness, tying together concepts from pruning, low-rankness, and Lipschitz theory.
- Empirical analysis covers diverse architectures and datasets, including FCN, convolutional and transformer families, and multiple attack settings

**Weaknesses:**

- The theory uses global operator norm–based Lipschitz bounds.
and The bounds rely on scale-normalized parameters (using ∥W∥_F) and strict (q, k, ε)-compressibility. Can this reflect practical training dynamics with normalization layers or adaptive scaling or deep non-linear networks.
- How does it position itself with other works exploring the same paradigm [1][2], As some works claim that some sparsity helps robustness
- The claim that compressibility fosters universal adversarial examples is intriguing but briefly demonstrated

[1] Lipschitz Constant Meets Condition Number: Learning Robust and Compact Deep Neural Networks
[2] Robust low-rank training via approximate orthonormal constraints

**Questions:**

- Why focus exclusively on structured compressibility (neuron/spectral)? Would unstructured or other forms behave differently?
- In Fig1,2  did not understand how to interpret these new directions? How were they visualized?
- The alignment equation notations could be explain bit better
- Line 325, how does Fig4 prove the dominant singular directions claim?

---

> ### Author Response · Authors · 2025-11-25
>
> We thank the reviewer for their thoughtful comments and appreciation of our work. We hope our answers address their concerns and lead to a positive reconsideration of their score. We are available for discussion regarding any remaining concerns.
>
> ---
>
> **W1: Scale-normalized parameters.** Our bound does **not** assume $||\mathbf{W}||_F = 1$; instead, it makes a vital distinction between **structure and scale**; we disentangle compressibility and Frobenius norm as two distinct contributors to the operator norm. Having separate terms for compressibility and Frobenius norm allows us to *analytically isolate* the effect of structured compressibility **without confounding by overall parameter scale**, and allows us to make novel predictions such as **appearance of UAEs** only in response to increased compressibility, but not $||\mathbf{W}||_F$. We only normalize $||\mathbf{W}||_F$ in some initial experiments as an intervention to **isolate the effect of compressibility**. In later, more realistic settings we instead use weight decay for norm control. We have now clarified this distinction in Section 4 **(L338-340)**.
>
> ---
> **W2: Comparison with previous work.** We see our results to be **complementary** to that of [1, 2] (i.e., Feng et al., 2025; Savostianova et al. 2023), even though our work's **scope extends beyond** these two studies. [1] finds that while *unstructured* sparsity is beneficial in low-medium levels, at high levels it reduces robustness due to a dramatic increase in the **condition number**. As we focus on the effects of _structured_ compressibility, our results are complementary with their negative findings [1], confirming the latter part of their results. Note that our $\ell_\infty$ operator norm bound also implies a positive role for unstructured sparsity, further aligning our results (Theorem 3.1). [2] observes that training with a low-rank factorized architecture leads to **increased condition number** and thus adversarial vulnerability. Again, this result is in agreement with our conclusions. However, we note that **our focus is more extensive**: (i) Our results are agnostic to the **source of compressibility** (therefore include but are not limited to the mechanisms pointed out in [1, 2]), (ii) We go beyond condition number by using a more **fine-grained definition of compressibility**, (iii) Our framework addresses **neuron compressibility** as well as spectral compressibility. We expanded Section 5 to explicitly incorporate this discussion **(L506-518)**.
>
> ---
> **W3: Universal adversarial examples (UAEs).** We welcome the reviewer's suggestion to expand on this promising direction. First, we **replicate** the results in Figure 7, which were demonstrated using MLPs, with ResNet18's trained under increasing row sparsification vs. increasing Frobenius norm. As in our original experiments, only **row sparsification leads to UAEs**, see [Figure 26](https://anonymous.4open.science/api/repo/11954/file/figure_26_uae.png?v=8acca860). To further probe this relationship, we conduct adversarial training with ResNet18s under increasing compressibility, either **with standard adversarial examples vs. UAEs**, and investigate resulting networks' parameters. Our results provide an exciting complement to our original results: **UAE training dramatically reduces spread of the dominant terms** ($\beta$) compared to standard adversarial training, implying that just as compressibility-induced vulnerable latent directions allow UAEs, **training against UAEs suppresses** such directions, as depicted in [Figure 27](https://anonymous.4open.science/api/repo/11954/file/figure_27_uae_training.png?v=f095cd38). We have added these results to our paper **(L423-426; Appendix D.11)**.
>
> ---
> **Q1: Why structured compressibility?** We focus on structured compressibility because it is the variant that **actually yields inference-time speedups** in standard hardware (Blalock et al, 2020), and because, to our knowledge, there is **no prior theory** characterizing its interaction with robustness. Note that our $\ell_\infty$ operator norm bound involves *unstructured compressibility* as well. We consider extending our research to 1- larger-scale structured compression such as layer or attention head pruning, and to 2- unstructured or semi-structured pruning as exciting **future directions**. We have incorporated this aspiration to our main text **(L535-537)**.
>
> ---
> ### Continued below

---

> ### Author Response · Authors · 2025-11-25
>
> ### Continued from above
> ---
> **Q3: How does Fig. 4 support dominant singular directions claim?** Figure 4 demonstrates our main results on a single hidden layer neural network $g(\mathbf{x}) = \mathbf{C} \phi(\mathbf{W}\mathbf{x})$. Increased nuclear norm regularization increases spectral compressibility, leading to a few large singular values $\sigma\_i$ (Figure 4, left). A perturbation $\mathbf{a}$ that would like to "exploit" this **magnitude increase** allowed by these large $\sigma\_i$'s would have to **"align"** with their associated *singular vectors* in the input space, $\mathbf{v}\_i$, i.e. $|\mathbf{v}_i^T\mathbf{a}/||\mathbf{a}||_2| \approx 1$. If decreased robustness under compressibility is due to **attacks exploiting these few potent directions** (e.g. $\mathbf{v}\_1, \mathbf{v}\_2, \mathbf{v}\_3$) as we argue, we should see $\sum^3\_{i=1}|\mathbf{v}\_i^T\frac{\mathbf{a}}{||\mathbf{a}||\_{2}}|$ increase under compressibility. This is exactly what the green line in Figure 4 (right) shows. And if under compressibility $\mathbf{a}$ inflate themselves by exploiting these potent directions, they should have **more dominant latent representations** (w.r.t. original image's representation). That is, $||\mathbf{z}\_{\mathrm{adv}} - \mathbf{z}||_2/||\mathbf{z}||_2$ should **increase** under compressibility. This is exactly what the blue line in Figure 4 (right) shows. We greatly expanded our discussion to provide a clearer understanding of this figure in the main text and Appendix **(L171-182; L339-341; Appendix D.9)**. Please refer to our response to Reviewer TtqN and our new Appendix D.9 for a more detailed explanation.
>
> ---
> **Q2: Figures 1 and 2.** Figure 2 is a **qualitative example** of what we described above, from the same experiment as Figure 4. It is created by a PCA of the input samples and representations, and plotting them over decision boundaries in the **input and representation** spaces. The main message is that, although standard and compressible models share the **same small budget** in input space, compressibility allows adversarial perturbations to attain a **much larger relative magnitude** in latent space and eventually **change the prediction**. Figure 1 is a schematization of this idea. We have greatly expanded the introduction of this figure and our motivating hypothesis to ensure a smoother onboarding **(L171-182)**.
>
> ----
> **Q4: Alignment notation.** We have revised our paper to **streamline the notation** and to provide a **stronger intuitive introduction** for this notion **(L228-248)**, as well as dedicating a **new subsection in the Appendix D.9** for a more detailed explanation. Please see our response to Reviewer 1FKt for further details.

---

> > ### Comment · Reviewer_Tszb · 2025-11-27
> >
> > I thank the authors for their detailed rebuttal and for providing further clarity and explanation about all the figures. I maintain my score

---

> > > ### Author Response · Authors · 2025-12-04
> > >
> > > We are pleased that our rebuttal helped address the reviewer's concerns and thank the reviewer again for their time and insightful feedback.

---

### Official Review · Reviewer_TtqN · 2025-11-01

**Soundness:** 3
**Presentation:** 3
**Contribution:** 2
**Rating:** 6
**Confidence:** 3

**Summary:**

This paper develops a framework to investigate the effect of structured sparsity on adversarial robustness through its effect on parameter norms and the network's Lipschitz constant. Compressibility can induce a set of highly sensitive directions in the representation space.

**Strengths:**

1. This paper is in general well-written and presents results clearly.
2. The motivating hypothesis is very interesting and described clearly in Figure 2.
3. Abundant numerical results are provided to testify the paper's theoretical results.

**Weaknesses:**

1. One central claim of this paper is that the compressibility may result to a few potent direction that increases the sensitivity to perturbations, and the adversarial attacks might exploit these directions. However, I cannot picture when and how the advesaries might be able to figure out these directions. Is the neural network and the compressibility totally white-box to the adversaries, which could hardly happen?
2. The evaluation of adversarially robustness of NN models seems to be dependent on the attack itself.

**Questions:**

1. Can you please offer a motivation example of how compressibility might be taken advantage of by adversaries? Especially how would adversaries figure out the "adversarial directions"?

---

> ### Author Response · Authors · 2025-11-25
>
> We thank the reviewer for their insightful questions and for their positive assessment of our work. We welcome the invitation to provide stronger intuition for our motivating hypotheses. We added a new section to our Appendix (D.9) that gives a **step-by-step**, visual account of how compressibility creates vulnerable directions and how **both white-box and black-box adversaries** exploit them. To respect the reviewer's time, we provide the core intuition here; details are available in Appendix D.9. We hope that our feedback and changes address the reviewer's concerns, and lead to a positive reconsideration of their score.
>
> ---
> **W1/Q1: How do adversaries figure out the vulnerable directions? Do we need white-box attacks for this?**
> We will now detail how these directions are easy to find, even for black box attacks. Let us consider a one hidden layer neural network under spectral compressibility as a running example. To visualize our exposition, we will use findings from actual experiments we conducted with a 400-unit MLP trained on CIFAR-10, under increasing nuclear norm regularization (NNR).
>
> Increased NNR leads to increased spectral compressibility, and to a few **dominant singular values** $\sigma_i$, as demonstrated in [Figure 19](https://anonymous.4open.science/api/repo/11954/file/figure_19_svs.png?v=33f7ed10), similar to Figure 4, left. Note that each strong singular value $\sigma_i$ has a direction in the input space associated with it (right singular vectors of the SVD, i.e. $\mathbf{v}_i$). It is very advantageous for an adversarial perturbation $\mathbf{a}$ to **"align"** with such a direction, i.e. $|\mathbf{v}_i^T\mathbf{a}/||\mathbf{a}||_2| \approx 1$, since if they do, **their magnitude** will be multiplied by this large $\sigma_i$ after they pass through the layer: $||\mathbf{W}\mathbf{a}||_2 \approx \sigma_i ||\mathbf{a}||_2$. Increasing their magnitude like this would allow them to dominate the original image $\mathbf{x}$ in the representation space. So despite a limited budget in the input space, they can eventually **change the prediction** of the model (see Figure 2).
>
> Let us now **visualize this "alignment"** for a more concrete understanding. We can compute  the alignment of an adversarial example with *the strongest* $I$ *singular directions* (SD) to understand which ones it is utilizing, i.e. $\rho\_i := |\mathbf{v}\_i^T \frac{\mathbf{a}}{||\mathbf{a}||\_2}| \in [0,1]$. See [Figure 20](https://anonymous.4open.science/api/repo/11954/file/figure_20_single_alignment.png?v=276e7a66) for a radar plot that shows this, with $I=20$. Now we can plot many samples in such a plot to see **if attacks concentrate** on a few leading singular directions under compressibility. [Figure 21](https://anonymous.4open.science/api/repo/11954/file/figure_21_samples_alignment.png?v=790f195c) shows that while in the baseline model **all 20 leading SDs are utilized**, under compressibility a **few strong SDs are overwhelmingly exploited** (as in Figure 4, right, blue line). Moreover, if adversaries are exploiting these potent directions, they should be **increasing their magnitude** vs. the original image in the representation space. That is, we should see that $||\mathbf{z}_\mathrm{adv} - \mathbf{z}||_2/||\mathbf{z}||_2$ is larger for more compressible models. The results in [Figure 22](https://anonymous.4open.science/api/repo/11954/file/figure_22_norm_increase.png?v=e3e56934) shows that this is exactly the case (similar to Figure 4, right, green line).
>
> ---
> ### Continued below

---

> ### Author Response · Authors · 2025-11-25
>
> ### Continued from above
> ---
>
> **Finding these directions with white box attacks.** Let's take **PGD** (Madry et al., 2018) as an example **white box** attack. Gradients point directly up the *steepest slope* in the input space. Directions $\mathbf{v}_i$ with large $\sigma_i$ are similar to **"ridges" in this landscape** with the steepest slopes; PGD converges to one such direction almost instantly. [Figure 23](https://anonymous.4open.science/api/repo/11954/file/figure_23_attacks.png?v=fc569510) (top) provides a *step by step visualization* from our experiment: PGD finds a potent direction in 6 steps.
>
> **Finding these directions with black box attacks.** Let's take **NES** (Ilyas et al. 2018) as an example **black box** attack. NES estimates gradients by *sampling random directions* in the *input space* and computing the changes in loss in each direction. While the loss landscape is diffuse in a standard model, in a compressed model, the **"ridges"** are so dominant that any random sample having even slight correlation with such a $\mathbf{v}_i$ yields a **high loss signal**. This guides the estimator **rapidly** toward the **vulnerability**. [Figure 23](https://anonymous.4open.science/api/repo/11954/file/figure_23_attacks.png?v=fc569510) (bottom) provides the visualization for *the same sample* with a NES attack. Although it is **not as efficient** due to lack of access to exact gradients, NES eventually converges to the **same direction** (in ~200 steps).
>
> Note that while we focused on $\ell_2$ attacks here, it is straightforward to apply a **similar analysis** to $\ell_\infty$ case, where the most vulnerable directions are rows $\mathbf{w}_i$ in $\mathbf{W}$ with the largest $||\mathbf{w}_i||_1$.
>
> ---
> **Q2: Evaluation-Attack Relationship.** [Figure 16](https://anonymous.4open.science/api/repo/11954/file/figure_16_alternative_attacks.png?v=e09db58b) replicates our original findings mentioned above (Figure 5, lower left; obtained with AutoPGD), with *FGSM*, *AutoCG*, *Square Attack*, and the composite *AutoAttack*. This confirms that our conclusions about compressibility and robustness are **not an artifact of a particular attack**. We add these results to Appendix D.6, which already include such replications under different budgets and norms. Critically, **Square Attack is a query-based black-box attack**; the fact that it replicates our AutoPGD results further confirms that these vulnerabilities are exposed even without direct gradient access.

---

### Author Response · Authors · 2025-11-25

We **thank all reviewers** for their constructive and insightful feedback. We are delighted that reviewers found our work **well-motivated**, **interesting**, **sound**, and **comprehensive**. We summarize the **main changes** we made for the rebuttal (see individual responses for more):

- **Clearer theoretical intuition**: We have modified our main paper to ensure a smooth onboarding and intuition to our theory and motivating hypotheses. We added a new section to the Appendix that provides a *step-by-step*, visualized account of how vulnerable directions emerge and how exactly these are exploited by (white box *and* black box) adversaries.
- **Novel experimental confirmation**: We added experiments that show (i) training against UAEs decreases the spread of dominant neurons, complementing our original high spread $\to$ UAEs results, (ii) regularizing interlayer alignment proves to be yet another effective intervention that our bound suggests, (iii) state-of-the-art adversarial pruning methods *implicitly* control the operator norms our theory identifies, consistent with our framework.
- **Streamlined exposition and context:** We modified our exposition for increased readability, and coupled this more accessible notation with stronger intuition for concepts such as interlayer alignment or top-$k$ spread. We also systematically clarified what makes our contributions unique with respect to prior literature.

All significant changes in the main text and in the Appendix are marked with blue. Any new figures mentioned in the comments are also provided as clickable anonymous links for convenience. We hope that our comments **addressed reviewers' concerns**, and in turn lead to a **positive reconsideration of their scores**. We are fully available for discussion should any concerns remain.

---

### Author Response · Authors · 2025-12-03
**Final Summary of Rebuttal and Discussions**

We sincerely thank all reviewers, Area and Program Chairs again for their time, effort, and valuable feedback. Below we summarize the final state of the discussion. We are pleased that our paper received mostly positive initial scores, with reviewers finding our work well-motivated, rigorous, and thorough. The reviewers also provided insightful suggestions and questions - as detailed below, we put significant effort into ensuring they are comprehensively addressed. In particular:

* In response to **Reviewer TtqN**'s request, we provided a step-by-step account of how white box and black box attacks can and do exploit vulnerable latent directions, with extensive new visualizations. We added a completely new section to the Appendix that includes this account in even more detail, and we streamlined the main paper’s writing to make these insights clearer. **These comprehensive additions were made following an already positive initial evaluation by the reviewer**.

* **Reviewer Tszb** requested further theoretical clarifications, more investigation into UAEs, and positioning of our work with respect to two particular previous works. We provided the clarifications and comparisons the reviewer asked for, and revised our paper accordingly. We provided additional results with UAEs that not only replicate our original results, but also support our proposed mechanism in a novel direction. **The reviewer expressed appreciation for our detailed response and maintained their positive evaluation**.

* **Reviewer LZxs** had questions regarding the novelty of our theory and motivating hypotheses. Although no specific prior work was cited, we highlighted the novelty and uniqueness of our contributions through additional literature analysis and a more detailed comparison with our work’s immediate precedents. We integrated this deeper discussion into our revised paper as well. **In their reply, the reviewer kindly expressed appreciation for our detailed rebuttal and stated that they were in the process of revising their assessment based on our responses**.

* **Reviewer 1FKt** asked for an improved intuition of how latent sensitive directions were exploited by adversaries, how these were propagated across layers, and how our work relates to adversarial pruning literature. We streamlined our discussions across all these points in the main paper and enriched our Appendix with additional content regarding interlayer alignment, including experiments that demonstrated it to be a viable intervention target for robust compression. Lastly, through empirical evaluations, we have shown that our theory is also useful in explaining the findings of two prominent adversarial pruning methods. **These clarifications and additional findings build on top of an already positive assessment by the reviewer.**

Overall, the constructive and insightful feedback provided by the reviewers strengthened our paper and its contributions. While the discussion period ended before most reviewers got a chance to interact with the paper further, we trust that the significant effort we put into thoroughly addressing the reviewers’ comments and enacting their suggestions will be taken into consideration in the final evaluation. We thank the reviewers and the Area Chair again for their time and consideration.

---

### Meta-Review · Area_Chair_66P4 · 2026-01-06

**Summary:**

This paper studies the interaction between structured compressibility (e.g., neuron-level sparsity and spectral compressibility) and adversarial robustness. The authors develop a principled theoretical framework connecting compressibility to operator norms, Lipschitz constants, and adversarial vulnerability, and derive robustness bounds that reveal how compressibility can induce highly sensitive directions in representation space. These theoretical insights are supported by extensive empirical evaluations across a wide range of datasets, architectures (including CNNs and transformers), and training regimes, including adversarial training and transfer learning.


The reviewers generally found the paper well-motivated, technically sound, and comprehensive. Initial concerns focused primarily on intuition, clarity, positioning with respect to prior work, and the practical relevance of the proposed mechanisms. The authors’ rebuttal and revisions substantially strengthened the paper by adding clearer intuition, new visualizations, additional theoretical clarification, and further experimental evidence, particularly regarding how adversaries exploit sensitive directions and how the framework relates to universal adversarial examples and prior robustness–compression literature.

**Reviewer Concerns:**

All substantive reviewer concerns have been successfully addressed in the rebuttal and subsequent revisions.


- Requests for clearer intuition on how compressibility creates vulnerable directions, and how adversaries (both white-box and black-box) can identify and exploit these directions, were thoroughly addressed with step-by-step explanations and new visualizations.
- Questions regarding the reliance on Lipschitz-based bounds, scale normalization, and practical relevance were clarified by explicitly disentangling structure and scale, and by adding experiments under more realistic training settings.
- Concerns about novelty and positioning relative to prior work on sparsity, low-rank models, and adversarial pruning were addressed through expanded discussion and comparisons.
- Requests for deeper investigation into universal adversarial examples and inter-layer alignment were met with additional experiments that further supported the proposed mechanisms.

Based on the rebuttal and discussion, there are no outstanding concerns.

**Reviewer Scores:**

Based on the strength of the rebuttal and the clarifications provided:

+ The three reviewers who initially provided positive evaluations are expected to maintain their scores of 6.
+ he reviewer who initially gave a 4 is likely to raise their score to 6, given that their main concerns were directly and convincingly addressed in the rebuttal.

---

### Decision · Program_Chairs · 2026-01-26

Accept (Poster)